# Vector-valued Representation is the Key: A Study on Disentanglement and Compositional Generalization

## Abstract

Disentanglement and compositional generalization are essential abilities for humans, as they enable rapid knowledge acquisition and generalization to new tasks. These abilities involve recognizing fundamental underlying concepts from observations and generating novel concept combinations. However, deep learning models often struggle with these capabilities. Numerous studies have proposed methods for disentangled representation learning, while recent research has also begun to address compositional generalization. Despite these advancements, the relationship between disentanglement and compositional generalization remains under-explored, with inconsistent findings reported in existing literature. In this paper, we analyze various prominent disentangled representation learning methods, examining their disentanglement and compositional generalization capabilities. Our study reveals a crucial insight: adopting vector-valued representations (using vectors rather than scalars to represent concepts) significantly enhances both disentanglement and compositional generalization performance. This insight resonates with findings from neuroscience research, which suggest that the brain encodes information through the collective activity of neuron populations, rather than relying on individual neurons. Motivated by this observation, we further propose a method to reform the scalar-valued disentanglement works ($\beta$-TCVAE and FactorVAE) to be vector-valued to increase both capabilities. We investigate the impact of the dimensions of vector-valued representation and one important question: whether better disentanglement indicates higher compositional generalization. In summary, our study establishes the feasibility of attaining both effective concept recognition and novel concept composition.

## 1 Introduction

Humans possess the ability to proficiently comprehend an extensive variety of abstract concepts, and to seamlessly generalize them to novel compositions of these concepts. This exceptional ability is suggested to be a crucial mechanism that enables humans to acquire knowledge and apply it to new contexts (Cole et al., 2013; Frankland & Greene, 2020; Ito et al., 2022). For example, humans can easily depict an unseen object using learned concepts such as color, shape, and texture. Language is generally considered a disentangled representation for visual observations, and can be recomposed to represent novel observations. Serving as a disentangled and computationally generalizable representation, language functions as a powerful tool that enables humans to comprehend the world, learn, and create knowledge. In a similar vein, it has been suggested that disentanglement (Bengio et al., 2013) and compositional generalization (Lake et al., 2017; Lake & Baroni, 2018) are fundamental missing ingredients for deep learning models to achieve human-like intelligence.

Toward this ambitious goal, the disentangled representation learning task has been proposed (Bengio et al., 2013) to uncover the underlying factors or concepts behind observations and to represent each factor with explicit representations. Various works have been proposed for this task, and one representative branch is VAE-based (Higgins et al., 2017a; Chen et al., 2018; Kim & Mnih, 2018). Two recent works, SAE (Leeb et al., 2022) and VCT (Yang et al., 2022), employ either an AdaIn-like structure or a Transformer to achieve disentanglement, respectively. VAE-based methods and SAE represent each factor with a single scalar, while VCT utilizes a vector, specifically a token, to repre-

sent each factor. While these methods achieve disentanglement, none evaluate their compositional generalization capabilities.

Compositional generalization has recently garnered attention. Montero et al. (2021) assessed compositional generalization in terms of image reconstruction and generation. A recent work (Xu et al., 2022) directly evaluates the compositional generalization ability of VAE-based methods, revealing that they demonstrate poor compositional generalization ability, and better disentanglement ability does not necessarily imply higher compositional generalization. Nonetheless, these studies focus exclusively on evaluating VAE-based disentangled representation learning methods; hence, it is essential to examine recent disentanglement approaches to elucidate the relationship between disentanglement and compositional generalization.

In this paper, we conduct a study on disentanglement and compositional generalization, unveiling a crucial insight: vector-valued representation is the key to facilitating both effective disentanglement and robust compositional generalization. By vector-valued representation, we mean employing a vector, rather than a scalar, to represent a factor. We investigate the latest vector-valued disentangled method, VCT (Yang et al., 2022), and discover that it can achieve both proficient disentanglement and potent compositional generalization. Inspired by this observation, we propose a method to vectorize the representations of two popular VAE-based methods ($\beta$-TCVAE (Chen et al., 2018) and FactorVAE (Kim & Mnih, 2018)) as well as SAE (Leeb et al., 2022). In addition to increasing the dimension of the latent vectors, we also need to reformulate the loss function and modify the architecture to meet the model's disentanglement requirements. The three vectorized methods exhibit enhanced compositional generalization, with an average increase of 51% and maintained disentanglement quality (some methods even show improvement) on Shapes3D compared to their scalar-valued counterparts. This observation is in conformity with the *population coding* in neuroscience. The brain encodes information in the population activity of neurons: *individual neurons count for little; it is population activity that matters* (Averbeck et al., 2006). Intuitively, scalar-valued representation (i.e., single neurons) provides limited information, whereas vector-valued representation enables the incorporation of more information for each concept. Consequently, we experiment with increasing the number of vector dimensions and observe that compositional generalization also improves. Analogous observations are also explored in (Xu et al., 2022), which demonstrates that increased bandwidth enhances compositional generalization in the Emergent Language Model. We further investigate the relationship between disentanglement and compositional generalization for vector-valued methods, observing a positive correlation between one of the compositional generalization metrics and disentanglement.

Our main contributions can be summarized as follows:

- We provide an important insight that vector-valued representation is one of the keys to both good disentanglement as well as strong compositional generalization.
- We introduce a vectorization technique to transform scalar-valued methods into vector-valued ones, thereby categorizing existing models into two groups: vector-valued and scalar-valued disentanglement methods.
- We conduct experiments to reveal the relation between disentanglement and compositional generalization for vector-valued methods: the compositional generalization classification metric positively correlates to the disentanglement, while the regression metric does not.

## 2 RELATED WORKS

### 2.1 DISENTANGLED REPRESENTATION LEARNING

Disentangled representation learning was first introduced by Bengio et al. (2013). The conventional disentangled representation implies that each scalar of the representation encodes a single independent factor, referred to as scalar-valued representation in this study. Several inductive biases have been proposed to achieve such scalar-valued disentanglement. For instance, VAE-based works impose constraints on the latent probabilistic distributions (Chen et al., 2018; Kim & Mnih, 2018; Higgins et al., 2017a; Burgess et al., 2018; Yang et al., 2021; Locatello et al., 2019). Schott et al. (2021); Montero et al. (2021) go beyond the VAE-based disentanglement studies, though they are still restricted to the scalar-valued representation. SAE (Leeb et al., 2022) proposes to adopt a

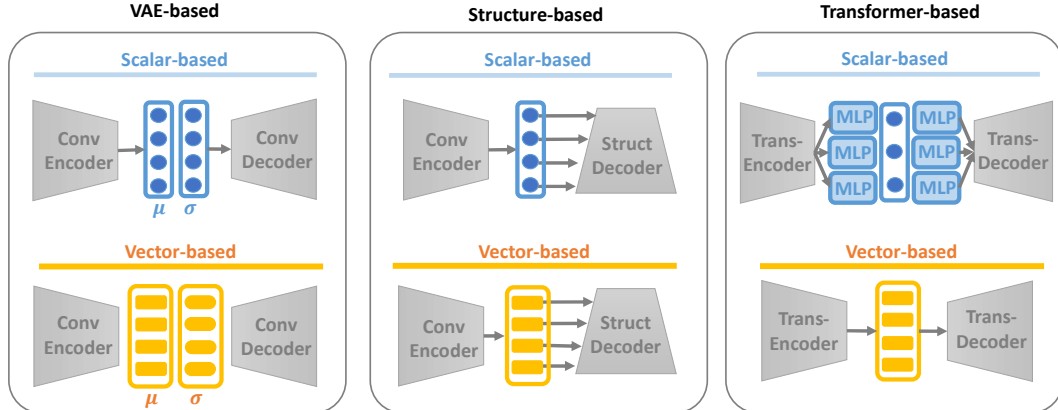

Figure 1: The unified illustration of scalar-valued and vector-valued disentanglement methods. For VAE-based and structure-based methods, we extend the scalar-valued output of the encoder into a vector-valued one, where each vector represents a factor. We reformulate the loss function of the VAE-based method in Section 3.3. A series of MLPs is employed to map each vector into a scalar for the transformer-based method.

StyleGAN generator-like architecture as the structure inductive bias. However, these models are, in general, only designed for disentangled representation learning, where compositional generalization is not considered. Singh et al. (2022) also emphasize the importance of vector-valued representation to compositional generalization. Few studies have focused on vector-valued representation in disentangled representation learning (Du et al., 2021). However, there is a considerable body of work (Tzelepis et al., 2021; Song et al., 2023; Wei et al., 2021) exploring how to identify the semantic directions in their latent space, and these directions are precisely represented by vectors. These representations are vector-valued for semantics. Although Yang et al. (2022) proposed a transformer-based model to learn vector-valued representation, to the best of our knowledge, no previous works have explored its compositional generalization ability.

## 2.2 COMPOSITIONAL GENERALIZATION

Without considering disentanglement, Zhao et al. (2018) study compositional generalization in generative models. The compositional generalization problem in disentangled representation learning was first studied in Esmaeili et al. (2019); Higgins et al. (2017b). However, only several specific forms of combinatorial generalizations have been explored, and the role of disentanglement on generalization is still not fully explored. Different from Montero et al. (2021), Xu et al. (2022); Schott et al. (2021) directly evaluates the compositional generalization and employs random train-test splits rather than manually selected splits. However, these three works are conducted on scalar-valued representation. Both studies find that the disentanglement of VAE-based methods is not correlated or even inversely associated with the compositional generalization. This inspires us to ask the question: is it possible that there exists a model possessing both abilities, and how can we train a model to achieve such a goal? There is line of works (Goyal et al., 2019; Träuble et al., 2023) emphasize the importance of vector-valued representations for generalization, inspiring us to explore that is it that the vector-valued representation also the key.

## 3 BACKGROUND: DISENTANGLEMENT MODELS

In this section, we introduce the disentanglement models used in this paper. We have chosen two widely-used VAE-based disentangling models: $\beta$-TCVAE (Chen et al., 2018) and FactorVAE (Kim & Mnih, 2018). In addition, we also consider two recently proposed disentanglement models: SAE (Leeb et al., 2022), a structure-based disentangling method, and VCT (Yang et al., 2022), a transformer-based approach.

### 3.1 VAE-BASED DISENTANGLEMENT

The disentangled representation learning assumes that the data $x$ is generated from a set of ground truth factors $\{f_i\}_{i=1}^N$. The goal of unsupervised disentangled representation learning is to a learn representation $z$ of data $x$ such that each unit $z_i$ is a function of a single factor $f_k$, where $1 \leq k \leq N$. VAE-based methods adopt total correlation as the regularization to encourage disentanglement.

Specifically, these two VAE-based methods decompose the total correlation from the KL regularization term of the vanilla VAE (Kingma & Welling, 2013). They thus penalize the total correlation with a hyper-parameter $\gamma$. The resulting loss function is:

$$\mathcal{L} = \mathbb{E}_{q(z|x)p(x)}\left[p_\theta(x|z)\right] - \mathbf{KL}(q_\phi(z|x)||p(z)) - \gamma\mathbf{KL}(q_\phi(z)||\prod_i q_\phi(z_i)), \tag{1}$$

where the last term represents the total correlation, and $p(z)$ is the prior distribution $\mathcal{N}(0, I)$. An encoder parameterized by $\phi$ models the conditional distribution $q_\phi(z|x)$. Conversely, a decoder parameterized by $\theta$ models the posterior $p_\theta(x|z)$. $\beta$-TCVAE and FactorVAE employ distinct methods to estimate the total correlation. Specifically, $\beta$-TCVAE utilizes the subsequent equation for estimation:

$$\mathbf{KL}(q_\phi(z)||\prod_i q_\phi(z_i)) = \mathbb{E}_{q_\phi(z)}[\log(q_\phi(z)) - \log(\prod_i q_\phi(z_i))]. \tag{2}$$

While the FactorVAE utilizes a discriminator $\mathcal{D}$ to approximate $q_\phi(z)$ and $\prod_i q_\phi(z_i)$, which is density-ratio trick. Consequently, the total correlation can be estimated as follows:

$$\mathbf{KL}(q_\phi(z)||\prod_i q_\phi(z_i)) = \mathbb{E}_{q_\phi(z)}[\log(\mathcal{D}(z)) - \log(1 - \mathcal{D}(z))], \tag{3}$$

where the discriminator $\mathcal{D}$ is learned by adversarial training simultaneously. The discriminator is trained to classify between samples originating from $q_\phi(z)$ and $q_\phi(\bar{z})$, where $\bar{z}$ is the representation permuted along dimension $i$.

### 3.2 STRUCTURE-BASED DISENTANGLEMENT

The VAE-based methods leverage the estimated total correlation as the regularization of VAE to achieve disentanglement. We also want to explore the relation between disentanglement and compositional generalization of the model without regularization. SAE introduces a structural decoder designed to learn a hierarchy of latent variables, enabling the factorization of encoded information without additional regularization. Therefore, we select this model in this paper. As depicted in Fig. 1, SAE employs an AdaIN-like structure to modulate the spatial feature for image reconstruction, which shares a similar architecture with StyleGAN (Karras et al., 2019). However, unlike StyleGAN, the injection layer maps an encoded scalar rather than a vector.

### 3.3 TRANSFORMER-BASED DISENTANGLEMENT

In conventional disentangled representation, a scalar encodes a single factor. This type of representation is referred to as scalar-valued representation in this paper and includes methods such as VAE-based approaches and SAE. Conversely, when a single factor is encoded in a vector, it is called a vector-valued representation. Since we have introduced three kinds of scalar-based methods above, we use vec-VCT as an example of a vector-valued method.

Vec-VCT employs stacked cross-attention layers to induce visual information from the image without self-attention between distinct concepts, effectively preventing information leakage across units. Moreover, a Concept Disentangling Loss is proposed to promote the mutual exclusivity among concept tokens. As illustrated in Fig. 1, vec-VCT learns a vector-valued disentangled representation. In the following sections, we introduce a *vectorization* method to transform scalar-valued representations into vector-valued ones, as illustrated in Fig. 1.

## 4 VECTORIZED REPRESENTATIONS

Given a sample $x$, the encoder of the VAE-based method produces scalar statistics, namely, mean $\mu_i$ and variance $\sigma_i$, where $i = 1, 2, \ldots, m$. The number of units of the representation is denoted by $m$.

To transform it into a vector-valued representation, for factor $i$, the encoder is modified to predict vectors $[\mu_{i1}, ..., \mu_{iD}]$ and $[\sigma_{i1}, ..., \sigma_{iD}]$ instead, where $D$ represents the dimension of each vector. As one unit encodes an individual semantic, the variance within each unit $i$ is set to be the same, i.e., $\sigma_{ij} = \sigma_{ik}, j \neq k$.

The loss function (Eq. 1) must also be adapted when applied to the vector-valued representation. As the first term represents the reconstruction loss and requires no modification, our focus lies on the last two terms. The KL divergence can be expressed as follows:

$$\mathbf{KL}(q_\phi(z|x)||p(z)) = -0.5(-\frac{1}{D}\sum_{ij}\mu_{ij}^2 - \sum_i(\sigma_i - \log \sigma_i) + m) \cdot D + C. \qquad (4)$$

Please refer to Appendix A for a detailed derivation. To ensure comparability with the vanilla VAE, the multiplier $D$ is ignored when calculating the loss. Additionally, the total correlation of $\beta$-TCVAE is modified. Specifically, since the total correlation of the vector-valued $\beta$-TCVAE with a shared variance within each unit is intractable, the total correlation is averaged along the dimension of the representation vector to approximate the total correlation of the vector-valued $\beta$-TCVAE:

$$\mathbf{KL}(q_\phi(z)||\prod_{i=1}^m q_\phi(z_i)) \approx \frac{1}{D}\sum_{j=1}^D \mathbf{KL}(q_\phi(z_{\cdot j})||\prod_{i=1}^m q_\phi(z_{ij})), \qquad (5)$$

where the marginal $q(z_{\cdot j})$ refers to the joint probability distribution of the $j$-th dimension of all the vector representations. Note that the average operation is used instead of the sum operation to make the value comparable to the original $\beta$-TCVAE. Combining Eq. 5 and Eq. 4, the loss function of $\beta$-TCVAE can be obtained. Unlike $\beta$-TCVAE, the discriminator $\mathcal{D}$ of FactorVAE must be modified to estimate the total correlation of a set of joint distributions. The discriminator $\mathcal{D}$ is extended to accept $z_{ij}$ as input, and the permutation is only performed on dimension $i$ during the training of the discriminator. For more details, please refer to Appendix B. Together with Eq. 3 and Eq. 4, the loss of vector-valued FactorVAE can be computed.

As mentioned earlier, to transform the structure-based method, SAE, into a vector-valued method, the encoded scalar must be replaced with an encoded vector of $D$ dimensions, as illustrated in Fig.1. Consequently, the scale and shift coefficients are predicted by a vector, which shares the same structure as the generator of StyleGAN (Karras et al., 2019). SAE utilizes a decoder structure where each layer encodes a unique factor. Consequently, for scalar-valued methods, each layer of the decoder is modulated by a single dimension. In contrast, in the vector-valued one, each layer is modulated by several dimensions. Since SAE is a regularization-free method, no modification of the loss function is required.

Since vec-VCT, a transformer-based method, is already a vector-valued method, it is modified into a scalar-valued method to facilitate the categorization of these models into two groups. Specifically, as illustrated in Fig. 1, different MLP layers are employed to map distinct vectors into scalars while preserving the independence of the learned vectors. Given that this modification does not impact the loss function, the loss function of vec-VCT is retained.

## 5 EXPERIMENT DESIGN

### 5.1 DATASET

In this study, the focus is on exploring disentanglement and compositional generalization ability. Two public datasets are commonly used in both disentangled representation learning (Yang et al., 2022; Leeb et al., 2022) and compositional generalization literature (Xu et al., 2022): Shapes3D (Kim & Mnih, 2018) and MPI3D-Real (MPI3D in short) (Gondal et al., 2019), are used in accordance with Xu et al. (2022).

**Data Splits** The aim of compositional generalization is to identify new combinations of previously encountered concepts in downstream tasks. To achieve this, the current study adheres to (Xu et al., 2022) for dividing the dataset into two portions: training and testing sets at a 1:9 ratio. Notably, the training set is smaller than those utilized in Montero et al. (2021) and Schott et al. (2021). The hyper-parameters for the models implemented in this study are derived from prior research. For

Table 1: A comparative analysis of disentanglement and compositional generalization between scalar-valued and vector-valued methods (mean $\pm$ std, higher is better) indicates that vector-valued methods surpass scalar-valued methods with a notably wider margin in compositional generalization performance. We use (640) to denote the scalar-valued methods with the same total dimension. For vec-VCT*, the parameter $D = 256$ aligns with the value utilized in Yang et al. (2022). The results for the $\beta$-VAE score and MIG can be found in Appendix E.

| Method | Shapes3D | | | | MPI3D | | | |
|--------|----------|----|----|-----|-------|----|----|-----|
| | FactorVAE | DCI | R2 | ACC | FactorVAE | DCI | R2 | ACC |
| *scalar-valued:* | | | | | | | | |
| FactorVAE | $0.83 \pm 0.06$ | $0.44 \pm 0.12$ | $0.46 \pm 0.18$ | $0.39 \pm 0.10$ | $0.31 \pm 0.04$ | $0.21 \pm 0.01$ | $0.30 \pm 0.02$ | $0.39 \pm 0.02$ |
| $\beta$-TCVAE | $0.83 \pm 0.10$ | $0.65 \pm 0.16$ | $0.45 \pm 0.15$ | $0.47 \pm 0.18$ | $0.44 \pm 0.05$ | $0.27 \pm 0.01$ | $0.32 \pm 0.03$ | $0.45 \pm 0.03$ |
| SAE | $0.98 \pm 0.04$ | $0.87 \pm 0.12$ | $0.72 \pm 0.05$ | $0.90 \pm 0.17$ | $0.71 \pm 0.04$ | $0.47 \pm 0.05$ | $0.55 \pm 0.07$ | $0.77 \pm 0.02$ |
| VCT | $0.95 \pm 0.05$ | $0.86 \pm 0.02$ | $0.56 \pm 0.24$ | $0.58 \pm 0.15$ | $0.72 \pm 0.04$ | $0.47 \pm 0.03$ | $0.39 \pm 0.13$ | $0.69 \pm 0.09$ |
| FactorVAE (640) | $0.77 \pm 0.05$ | $0.56 \pm 0.12$ | $0.77 \pm 0.10$ | $0.65 \pm 0.10$ | $0.46 \pm 0.03$ | $0.43 \pm 0.01$ | $0.37 \pm 0.03$ | $0.49 \pm 0.02$ |
| $\beta$-TCVAE (640) | $0.81 \pm 0.11$ | $0.74 \pm 0.10$ | $0.59 \pm 0.15$ | $0.76 \pm 0.18$ | $0.43 \pm 0.02$ | $0.39 \pm 0.01$ | $0.35 \pm 0.02$ | $0.44 \pm 0.02$ |
| *vector-valued:* | | | | | | | | |
| vec-FactorVAE | $0.93 \pm 0.06$ | $0.55 \pm 0.11$ | $0.88 \pm 0.05$ | $0.96 \pm 0.02$ | $0.38 \pm 0.06$ | $0.16 \pm 0.05$ | $0.53 \pm 0.02$ | $0.71 \pm 0.01$ |
| vec-$\beta$-TCVAE | $0.82 \pm 0.08$ | $0.31 \pm 0.08$ | $0.87 \pm 0.05$ | $0.98 \pm 0.01$ | $0.42 \pm 0.06$ | $0.11 \pm 0.03$ | $0.67 \pm 0.02$ | $0.78 \pm 0.01$ |
| vec-SAE | $0.89 \pm 0.08$ | $0.63 \pm 0.06$ | $0.95 \pm 0.01$ | $0.98 \pm 0.01$ | $0.62 \pm 0.08$ | $0.33 \pm 0.09$ | $0.87 \pm 0.03$ | $0.88 \pm 0.01$ |
| vec-VCT | $0.98 \pm 0.04$ | $0.85 \pm 0.06$ | $0.91 \pm 0.10$ | $0.80 \pm 0.09$ | $0.70 \pm 0.06$ | $0.48 \pm 0.04$ | $0.70 \pm 0.07$ | $0.77 \pm 0.02$ |
| vec-VCT* | $0.97 \pm 0.04$ | $0.89 \pm 0.02$ | $0.99 \pm 0.02$ | $0.90 \pm 0.03$ | $0.66 \pm 0.03$ | $0.45 \pm 0.06$ | $0.85 \pm 0.07$ | $0.78 \pm 0.02$ |

$\beta$-TCVAE and FactorVAE, the `disentanglement_lib` implementation is employed (Locatello et al., 2019). Furthermore, the regularization strength $\gamma$ is assigned a value of 10, consistent with Kim & Mnih (2018) and Chen et al. (2018). In the case of SAE, the SAE-12 model architecture is adopted as per Leeb et al. (2022). A training batch size of 32 is employed, and the Adam optimizer is used with a learning rate of $10^{-4}$. For additional details, please consult Appendix D.

## 5.2 EVALUATION METRICS

**Disentanglement Evaluation** In this study, we adhere to Xu et al. (2022) by concentrating on the testing set performance, which demonstrates the model's capacity to disentangle unobserved factor combinations. We adopt their methodology of employing 3 random seeds for dataset splitting. We conform to (Locatello et al., 2019) by executing our experiments with 5 random seeds for each splitting. This results in $15 = 5 \times 3$ runs for each method on each dataset. In line with Yang et al. (2022), our experiments utilize four widely recognized metrics: the FactorVAE score(Kim & Mnih, 2018), the DCI (Eastwood & Williams, 2018), the $\beta$-VAE score (Higgins et al., 2017a), and MIG (Chen et al., 2018). We follow Du et al. (2021); Yang et al. (2022) to perform PCA as post-processing on the representation and evaluate the performance of vector-valued representations.

**Compositional Generalization Evaluation** Xu et al. (2022) assesses compositional generalization by investigating the ease with which a simple model can predict the ground truth of factors in novel combinations. In line with Xu et al. (2022), we train a straightforward classifier and regressor on the learned representation using $N_{label} = 500$ labeled data points. Specifically, we employ a ridge regression model for regression tasks and logistic regression for classification tasks. Consequently, the evaluation metrics include the $R^2$ score (R2) and classification accuracy (ACC).

## 6 KEY STUDY AND RESULTS

### 6.1 VECTOR-VALUED REPRESENTATION CAN POSSES BOTH DISENTANGLING AND COMPOSITIONAL GENERALIZATION

In this section, we conduct a comparative analysis of the disentanglement and compositional generalization capabilities of various models employing scalar-valued and vector-valued representations. Tab. 1 illustrates the experimental results. We emphasize the following key observations:

**Vector-valued representation** Upon comparing vector-valued and scalar-valued methods with identical inductive biases, the compositional generalization performance of vector-valued approaches consistently outperforms their scalar-valued counterparts, irrespective of the method employed. Given that the vector-valued method retains the inductive bias, only a minor reduction in disentanglement performance is observed. In certain cases, such as FactorVAE on Shapes3D, the vector-valued FactorVAE even demonstrates enhanced performance. However, due to the utilization of an

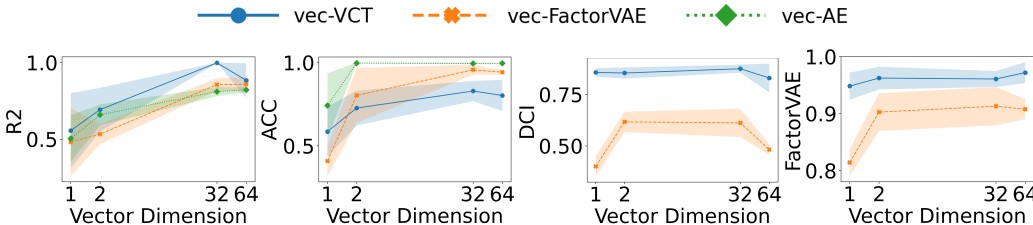

Figure 2: An analysis of compositional generalization and disentanglement performance with respect to vector size is conducted on the Shapes3D dataset., employing vec-VCT, vec-FactorVAE, and vec-AE models with a regularization strength parameter of $\gamma = 10$. Our findings reveal a positive correlation between the generalization metrics, R2 and ACC, and vector size (vector dimension).

approximate total correlation, a notable decline in the performance of vec-$\beta$-TCVAE is evident. The vec-VCT method achieves state-of-the-art performance in both disentanglement and compositional generalization domains.

**Implications** Our findings contrast with those presented by Xu et al. (2022), who argue that superior disentangled representations yield inferior compositional generalization. Their research focuses on scalar-valued disentangling methods. In contrast, our study demonstrates that, in the context of vector-valued methods with some certain inductive bias, enhanced disentanglement does not compromise compositional generalization performance. By increasing the bottleneck bandwidth, vector-valued methods outperform their scalar-valued counterparts in terms of compositional generalization.

## 6.2 LARGE VECTOR SIZE RESULTS IN BETTER PERFORMANCE FOR BOTH ABILITIES

In this section, we examine vector-valued representations across a range of vector sizes. Intuitively, an increased vector size equates to a wider bottleneck bandwidth. We implement vec-FactorVAE and vec-VCT, while also training a vanilla AE as a baseline for comparison. We assess the performance of these models using distinct vector sizes, specifically $1, 2, 32, 64$.

The findings are presented in Fig. 2. As vector size expands, generalization performance exhibits consistent improvement across all models. However, a slight decrease in performance is observed when the vector size surpasses 32. Conversely, disentanglement inductive bias exerts a detrimental impact on classification, yet remains favorable for regression performance. In terms of disentanglement performance, a larger vector size also results in superior disentangling efficacy.

**Implications** Elevating the bottleneck bandwidth results in improved generalization performance. Nonetheless, an increased vector size introduces additional complexity to the latent space, leading to a reduction in performance when the vector size surpasses 32. Despite the models demonstrating analogous behavior in disentanglement and generalization throughout this experiment, the correlation between these two competencies remains ambiguous. For instance, vec-FactorVAE and vec-VCT exhibit distinct patterns between $R^2$ and ACC in comparison to AE.

## 6.3 RELATION BETWEEN DISENTANGLEMENT AND COMPOSITIONAL GENERALIZATION

In the following section, we delve deeper into the interrelationship between disentanglement and compositional generalization performance in the context of vector-valued representations. We train vec-FactorVAE and vec-$\beta$-TCVAE models with varying regularization strengths. We adhere to the guidelines set forth by Kim & Mnih (2018) and Chen et al. (2018), and assign regularization strengths of $5, 10, 20$ to both models. Finally, we evaluate the trained vec-$\beta$-TCVAE, vec-FactorVAE, and vec-VCT models, and determine the correlation between disentanglement and compositional generalization.

The findings from our experiments are presented in Fig. 3. Notably, an increase in regularization strength leads to a decline in disentanglement performance for vec-FactorVAE, while it enhances performance for vec-$\beta$-TCVAE. Although compositional generalization performance exhibits a similar pattern, its impact on the regression metric remains marginal. Moreover, increasing the regular-

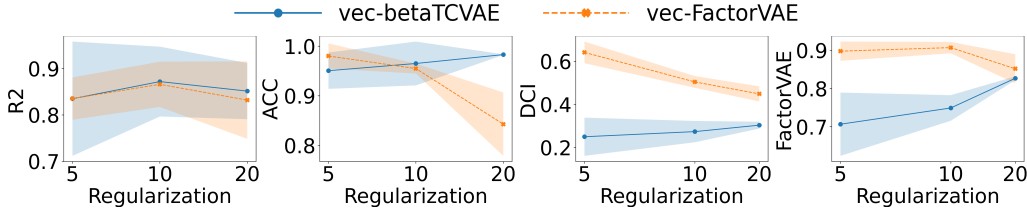

Figure 3: An analysis of compositional generalization and disentanglement performance with respect to regularization strength is conducted on the Shapes3D dataset. We evaluate vec-betaTCVAE and vec-FactorVAE, both employing a vector size of $D = 64$. These models are assessed using a range of regularization strengths, specifically $\gamma$ values within the set $5, 10, 20$.

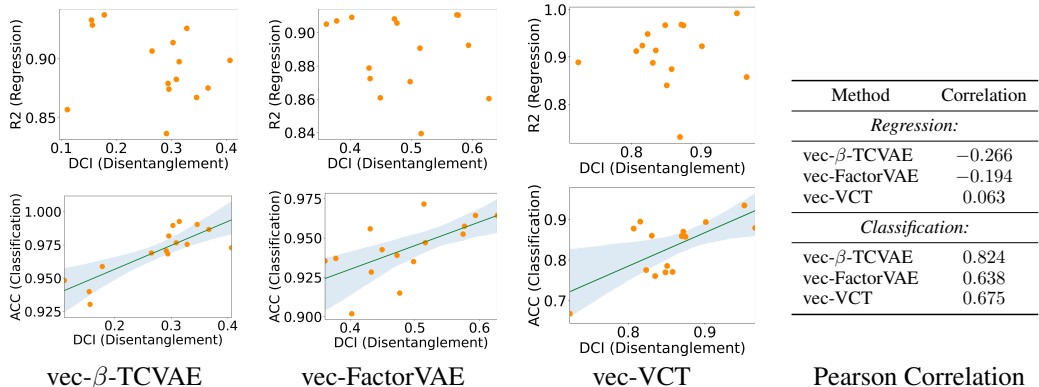

Figure 4: A correlation analysis is performed to investigate the interplay between compositional generalization and disentanglement in the context of the Shapes3D dataset, utilizing a vector size of $D = 64$ and a regularization strength parameter of $\gamma = 10$. The calculated Pearson correlation coefficient is subsequently displayed in the tabulated results. Notably, orange data points signify instances where the same vector-valued method has been trained using distinct random seeds.

ization strength for $\beta$-TCVAE results in a reduction of performance variance, aside from the changes in performance. Fig. 4 displays the performance of models trained with consistent regularization strengths, revealing a positive correlation between classification and disentanglement performance, with no significant correlation observed for regression. To further corroborate this relationship, we calculate the Pearson correlation coefficient. As shown in the table within Fig. 4, classification displays a positive Pearson correlation coefficient, while regression presents zero or slightly negative correlations. This finding bolsters our conclusions. We postulate that the performance deterioration in vec-FactorVAE arises due to an excessively large regularization strength in FactorVAE, causing a marked performance drop. This observation is also supported by Fig. 5 from Kim & Mnih (2018). Additional experiments with $\gamma \leq 5$ are provided in Appendix F.

**Implications** In the context of vector-valued representation, we observe a strong positive correlation between the metric of classification and disentanglement, while the metric of regression does not show such a strong correlation. We speculate that the underlying reason might be that regression primarily occurs in factors with multiple values, such as azimuth with 15 values on Shapes3D (for classification, such as scale with 3 values). This leads to the representation of this factor will encode more information. The size of the vector dimensions significantly impacts R2 in such cases.

## 6.4 Discussion on Estimation of the vector-valued TC

We have attempted the following different ways for estimation of the vector-valued TC. Note that estimating the total correlation (TC) in vector-valued representations is inherently a difficult problem. The details of these methods are described in Appendix C, and the experimental results demonstrate that our method performs better in practice than these alternatives: $(i)$ The method that directly derive the total correlation from multidimensional Gaussian distribution, referred to as "vec-betaTCVAE Sum". $(ii)$ During each training step, we randomly select a variable for each representation vector to calculate TC, referred to as "vec-betaTCVAE Rand". $(iii)$ We calculate TC as the average of multiple runs of the method $(ii)$, referred to as "vec-betaTCVAE Rand Sum". The

Table 2: The comparison with different total correlation estimation methods for vector-valued $\beta$-TCVAE on Shapes3D.

| Method | Factor | DCI | $R^2$ | ACC |
|---|---|---|---|---|
| vec-$\beta$-TCVAE Sum | $0.58 \pm 0.14$ | $0.22 \pm 0.09$ | $0.79 \pm 0.09$ | $0.73 \pm 0.21$ |
| vec-$\beta$-TCVAE Rand | $0.79 \pm 0.08$ | $0.31 \pm 0.07$ | $0.88 \pm 0.05$ | $0.96 \pm 0.02$ |
| vec-$\beta$-TCVAE Rand Sum | $0.52 \pm 0.09$ | $0.32 \pm 0.10$ | $0.70 \pm 0.07$ | $0.51 \pm 0.04$ |
| vec-$\beta$-TCVAE (Ours) | $0.82 \pm 0.08$ | $0.31 \pm 0.08$ | $0.87 \pm 0.05$ | $0.98 \pm 0.01$ |

performance of these methods is shown in Tab. 2. Based on the results, our current vectorization method outperforms the other methods.

## 6.5 SANITY CHECK ON EXPERIMENTS

Given that vector-valued methods utilize $D$ times more dimensions, it is essential to investigate whether their performance benefits arise from the increased dimensionality within the latent space. To examine this, we adjust the total dimensionality of scalar-valued methods to match that of vector-valued counterparts for $\beta$-TCVAE and FactorVAE. However, for SAE and VCT, since more latent units would result in a larger number of parameters for these models, leading to an unfair comparison, we do not conduct this experiment on SAE and VCT models. As demonstrated in Tab. 1, vector-valued methods continue to exhibit a significant performance advantage over scalar-valued approaches. Another important question to address is whether directly increasing feature dimensions affects performance metrics. To investigate this, we design experiments that evaluate various artificial representations, aiming to rule out the possibility that the vector-valued model's gains are solely attributable to the direct increase in feature dimensions. For this part of sanity checks, please refer to Appendix I.

## 7 CONCLUSIONS AND DISCUSSIONS

In this paper, we present a unified framework for vector-valued and scalar-valued disentanglement methods. Specifically, we adapt scalar-valued disentanglement techniques ($\beta$-TCVAE, FactorVAE, and SAE) to extend their vector-valued counterparts, while also converting the vector-valued method (VCT) into a scalar-valued version. We evaluate these disentangled representation learning methods concerning disentanglement and compositional generalization capabilities. Our investigation unveils a notable finding: employing vector-valued representations (using a vector instead of a scalar to represent a concept) is pivotal for achieving both effective disentanglement and robust compositional generalization. This insight underscores the significance of vector-valued disentangled representations. We observe that augmenting the dimensionality of vector-valued representations leads to improved compositional generalization. By concentrating on vector-valued disentanglement, we reassess the relationship between disentanglement and compositional generalization, discovering a positive correlation between classification and disentanglement for vector-valued approaches. Our study aims to stimulate further research on compositional generalization and vector-valued disentanglement techniques. Promising future directions encompass the examination of more complex, large-scale natural datasets with numerous factors and the exploration of the relationship between disentanglement in real-world, large-scale natural datasets and compositional generalization. Due to the page limitation, we discuss the limitation of our work in Appendix H. To vectorize a method not covered in our paper, we suggest the following systematic set of principles: $(i)$ Identify the basic structure of the model that learns a single factor. Then, extend this basic structure from one dimension to multiple dimensions. $(ii)$ Confirm that the inductive bias for disentanglement remains valid after the extension. $(iii)$ If the inductive bias still holds after the extension, then we have successfully vectorized the method. If it does not, we need to explore how to modify the inductive bias so that it remains valid for vector representations.

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

# A  MORE DETAILS OF LOSS FUNCTION

## A.1  KL DIVERGENCE FOR VECTOR-VALUED VAE METHODS

**Theorem 1** *Let $x$ be an $n$-dimensional random vector. Assume $x$ is sampled from either one of the two multivariate normal distributions $p$ or $q$, which are specified by mean vectors $\mu_p, \mu_q \in \mathbb{R}^n$, and covariance matrices $\Sigma_p, \Sigma_q \in \mathbb{R}^{n \times n}$, respectively, as $x \sim p(\mu_p, \Sigma_p)$ and $x \sim p(\mu_q, \Sigma_q)$. Then, the Kullback-Leibler (KL) divergence of $p$ from $q$ is given by*

$$\mathbf{KL}(p(x)||q(x)) = \frac{1}{2}\left[(\mu_q - \mu_p)^T \Sigma_q^{-1}(\mu_q - \mu_p) + tr(\Sigma_q^{-1}\Sigma_p) - \log\frac{|\Sigma_p|}{|\Sigma_q|} - n\right]. \tag{6}$$

As introduced in Section 4.1 of the main paper, we use a spherical Gaussian for each latent representation unit in the vectorized VAE-based model. For the $i$-th representation unit of VAE, we use a mean vector $\mu_i$ and a scalar $\sigma_i$ to characterize the spherical Gaussian $q(z_i|x) = \mathcal{N}(\mu_i, \sigma_i I)$. Therefore, according to the theorem above, the KL divergence between the posterior and prior is as follows:

$$\begin{aligned}
\mathbf{KL}(q(z|x)||p(z)) &= \sum_i \mathbb{E}_{q(z_i|x)}(\log q(z_i|x) - \log p(z_i)) \\
&= \sum_i 0.5\left(\sum_j \mu_{ij}^2 + D\sigma_i - D\log\sigma_i - D\right) \\
&= 0.5\left(1/D \sum_i \sum_j \mu_{ij}^2 + \sum_i \sigma_i - \sum_i \log\sigma_i - m\right)D \\
&= 0.5\left(1/D \sum_{ij} \mu_{ij}^2 + \sum_i(\sigma_i - \log\sigma_i) - m\right)D
\end{aligned} \tag{7}$$

where $m$ is the number of latent representation units of the VAE-based model. We use a spherical Gaussian to characterize each unit. And $\mu_{ij}$ indicates the $j$-th entry of the mean vector $\mu_i$.

## A.2  THE TOTAL CORRELATION FOR VEC-$\beta$-TCVAE

Recall that $\beta$-TCVAE utilizes a sampling method to estimate the total correlation. Specifically, in order to estimate the expectation of $\log q(z)$, we first sample a mini-batch of images: $\{x^1, x^2, \ldots, x^M\}$, and estimate $\mathbb{E}_{q(z)}[q(z)]$ as follows:

$$\mathbb{E}_{q(z)}[q(z)] \approx \frac{1}{M}\sum_{k=1}^{M}\left[\log\sum_{l=1}^{M}\exp(E(z(x_l)|x_k)) - \log MK)\right], \tag{8}$$

where $E(z(x^l)|x^k) = \log q(z(x^l)|x^k)$ is the log density of the distribution $q(z(x^l)|x^k)$, and $K$ is the number of samples of the dataset. We use $z(x^l)$ to indicate the representation derived by sample $x^l$. The total correlation can be estimated by the following equation:

$$\mathbf{KL}\left(q(z)||\Pi_i q(z_i)\right) = \mathbb{E}_{q(z)}\left[\log q(z) - \sum_i \log q(z_i)\right]. \tag{9}$$

where $z_i$ denotes the $i$-th latent representation unit.

For the vector-valued case, we know that it is important to calculate the log density of the distribution to estimate the total correlation. The log density of spherical Gaussian can be computed as follows:

$$E(z_i|x^k) = \sum_{j=1}^{D} E(z_{ij}|x^k), \tag{10}$$

which indicates that $E(z_i|x^k)$ is a large negative scalar, since $E(z_{ij}|x^k) < 0$, the value of the probability $q(z_i|x^k) = \exp(E(z_i|x^k))$ is expected to be very close to 0, which requires a high degree of precision. However, the limited precision of GPUs produces relatively large truncation errors when dealing with these small numerical values. Therefore, the original estimator is no longer fit for vector-valued $\beta$-TCVAE, and the total correlation is intractable in this way. Therefore, we use the following way to substitute the estimated total correlation instead.

We constrain a necessary condition of the independence of the marginal distribution $q(z_i)$. Specifically, the total correlation is computed by the same entry of different representation units, i.e., $z_{ij}$ and $z_{kj}$ are independent, for $i \neq k$. We then can derive a new equation for the total correlation:

$$\mathcal{L}_{\beta\text{-TCVAE}} = \sum_j (\mathbf{KL}\left(q(z_j)||\Pi_i q(z_{ij}))\right), \tag{11}$$

where $\mathcal{L}_{\beta\text{-TCVAE}}$ is the regularization term of the vec-$\beta$-TCVAE loss function, which plays the role of total correlation $\mathbf{KL}(q(z)||\Pi_i q(z_i))$. Similarly, we also normalize it by dividing the vector dimension $D$, and then the value of the total correlation regularization is comparable to the original one of $\beta$-TCVAE.

From Eq.11, we see that each term in the summation is the total correlation of one-dimensional Gaussians and there is no need for compute Eq.10 for each unit. Therefore, the abovementioned issue does not exist. Since the regularization constraint of vec-$\beta$-TCVAE is only a necessary condition for the original model, the disentanglement performance of vec-$\beta$-TCVAE is relatively poor. In the following, we provide the theorem and proof of the necessity. For simplicity, we use two random vectors as an example.

**Theorem 2** *If random vectors* $\mathbf{z}_1 = \{z_{11}, \ldots, z_{1D}\}$ *and* $\mathbf{z}_2 = \{z_{21}, \ldots, z_{2D}\}$ *are independent, the corresponding random variables* $z_{1i}$ *and* $z_{2i}$ *are independent, where* $1 \leq i \leq D$.

*Proof*: Given that random vectors $\mathbf{z}_1$ and $\mathbf{z}_2$ are independent, their joint probability distribution can be expressed as the product of their marginal probability distributions:

$$p(\mathbf{z}_1, \mathbf{z}_2) = p(\mathbf{z}_1)p(\mathbf{z}_2). \tag{12}$$

Now let's consider any two random variables: $z_{1i}$ from $\mathbf{z}_1$ and $z_{2i}$ from $\mathbf{z}_2$. We want to show that $p(z_{1i}, z_{2i}) = p(z_{1i})p(z_{2i})$. First, we can write the joint probability distribution of $z_{1i}$ and $z_{2i}$ using the joint distribution of the vectors $\mathbf{z}_1$ and $\mathbf{z}_2$:

$$p(z_{1i}, z_{2i}) = \int_{\forall z_{1k} \neq z_{1i}, \forall z_{2k} \neq z_{2i}} p(\mathbf{z}_1, \mathbf{z}_2) dz_{1k} dz_{2k}. \tag{13}$$

Since $\mathbf{z}_1$ and $\mathbf{z}_2$ are independent, we can substitute their product of marginals:

$$p(z_{1i}, z_{2i}) = \int_{\forall z_{1k} \neq z_{1i}} \int_{\forall z_{2k} \neq z_{2i}} p(\mathbf{z}_1)p(\mathbf{z}_2) dz_{1k} dz_{2k}. \tag{14}$$

Notice that the summation is not affected by the values of $z_{1i}$ and $z_{2i}$. Therefore, we can separate the summation into two parts, one for each vector:

$$p(z_{1i}, z_{2i}) = \left( \int_{\forall z_{1k} \neq z_{1i}} p(\mathbf{z}_1) dz_{1k} \right) \left( \int_{\forall z_{2k} \neq z_{2i}} p(\mathbf{z}_2) dz_{2k} \right). \tag{15}$$

By definition, the summation of all the marginal probabilities for each vector is equal to the marginal probability of $z_{1i}$ and $z_{2i}$:

$$p(z_{1i}, z_{2i}) = p(z_{1i})p(z_{2i}). \tag{16}$$

Thus, if random vectors $\mathbf{z}_1$ and $\mathbf{z}_2$ are independent, then the corresponding random variables $z_{1i}$ and $z_{2i}$ are also independent.

# B  MORE DETAILS OF VEC-FACTOR VAE

For vec-FactorVAE, a factor is encoded by a vector $\{z_i, i = 1, \ldots, m\}$. We estimate the vector-valued total correlation of the representations using a method similar to FactorVAE, i.e., $\mathbf{KL}(q_\phi(z)|| \prod_i q_\phi(z_i))$. Both $q_\phi(z)$ and $q_\phi(z_i)$ are distributions of random vectors. If we can sample from these two distributions, we can estimate and optimize the total correlation by using the density ratio trick.

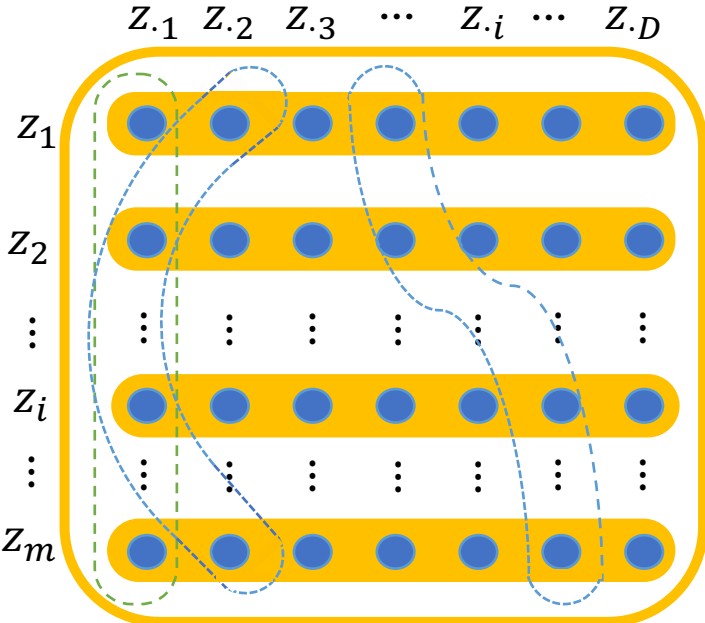

Figure 5: The illustration of the estimation of total correlation on the representation matrix. There are $m$ representation vectors, each with a dimension of $D$.

Following FactorVAE, we can sample a batch $z$ from $q_\phi(z)$ by sampling from data distribution and encoding. Alternatively, we can re-sample from $q_\phi(z_i)$ by permuting a batch of each $z_i$ in a different order for each $i$. we denote the permutated representation as $\bar{z}$. As long as the batch is large enough, the distribution of $\bar{z}$ will closely approximate $\prod_i q_\phi(z_i)$. According to density ratio trick, if we have a network $\mathcal{D}$ that can predict the probability that input sample $z$ is sampled from $q_\phi(z_i)$. We can then infer the probability that input $z$ is sampled from $\prod_i q_\phi(z_i)$ to be $1 - \mathcal{D}$. The total correlation can be formulated as

$$\mathbf{KL}(q_\phi(z)||\prod_i q_\phi(z_i)) = \mathbb{E}_{q_\phi(z)}\left[\frac{\mathcal{D}(z)}{1 - \mathcal{D}(z)}\right], \tag{17}$$

The network $\mathcal{D}$ is trained to classify between samples from $q_\phi(z)$ and $\prod_i q_\phi(z_i)$, thus learning to approximate the density ratio needed for estimating TC.

## C  THE TOTAL CORRELATION FOR VEC-$\beta$-TCVAE

**vec-betaTCVAE Sum.** According to $\beta$-VAE the expectation of the log density for $M$ samples can be estimated by the following:

$$\mathbb{E}_{q(z)}[\log q(z)] \approx \frac{1}{M}\sum_{l=1}^{M}\left[\log\sum_{k=1}^{M} q(z^l|x^k) - C\right], \tag{18}$$

where $q(z^l|x^k)$ represents the distribution of the representation given the $k$-th data sample $x^k$ in the batch as the condition. $z^l$ denotes the encoded representation of the $l$-th data sample. The log density of the multivariate normal distributions for each representation vector $z_i$ can be formulated as follows:

$$\log q(z_i^l|x^k) = -0.5 \cdot [(\mu_i(x^k) - z_i^l)^T\Sigma_i^{-1}(\mu_i(x^k) - z_i^l) + 0.5 \cdot \log|\Sigma_i| + C_1]. \tag{19}$$

The log density of the multivariate normal distributions for full representation vector $z$ can be formulated as follows:

$$\log q(z^l|x^k) = -0.5 \cdot [(\mu(x^k) - z^l)^T\Sigma^{-1}(\mu(x^k) - z^l) + 0.5 \cdot \log|\Sigma| + C_2]. \tag{20}$$

The total correlation can be estimated as follows:

$$
\begin{aligned}
TC(z) \quad &= \mathbb{E}_{q(z)}[\log q(z)] - \mathbb{E}_{q(z)}\left[\log \prod_i q_\phi(z_i)\right] \\
&= \frac{1}{M}\sum_{l=1}^{M}\left[\log\sum_{k=1}^{M}\exp\log q(z^l|x^k)\right] - \frac{1}{M}\sum_{l=1}^{M}\left[\sum_i\log\sum_{k=1}^{M}\exp\log q(z_i^l|x^k)\right].
\end{aligned}
\tag{21}
$$

The representation of the vector-valued methods can be rearranged into a matrix, with each row corresponding to a representation vector. As depicted in Figure 5, **vec-betaTCVAE** computes the total correlation on the column variables (indicated by the green dotted line). However, considering that the variables in different columns should also be taken into account, we examine a design that computes total correlation on randomly selected variables (one variable per row, indicated by the blue dotted line), which is referred to as **vec-betaTCVAE Rand**. The sampling of variables per training step may be insufficient. We can also have multiple runs of randomly selecting variables and computing the average TC, which is referred to as **vec-betaTCVAE Rand Sum**.

## D  MORE IMPLEMENTATION DETAILS

**Model architectures**. In our experiments, we follow (Ren et al., 2021)[1] and use the convolutional auto-encoder architectures for VAE-based methods, which is presented in Table 3 and 4. For vec-VCT[2], we use $m$ MLPs to reduce the dimension of $m$ concept tokens from 256 to $D$, which presented in Table 5 (a). For vec-VCT*, we do not use these MLPs to maintain the dimension as $D = 256$. We follow (Leeb et al., 2022) to use the same architecture of SAE[3], and change the MLP in the encoder and Str-Tfm layer to produce the vector-valued representation, as shown in Table 5 (b) and (c).

Table 3: Encoder $\mathcal{E}$ used in vec-VAE-based methods. We use $m$ to denote the number of latent representation units, and $D$ to denote the vector dimension. We set $D = 1$ for scalar based methods.

| |
|---|
| Conv $7 \times 7 \times 3 \times 64$, stride $= 1$ |
| ReLu |
| Conv $4 \times 4 \times 64 \times 128$, stride $= 2$ |
| ReLu |
| Conv $4 \times 4 \times 128 \times 256$, stride $= 2$ |
| ReLu |
| Conv $4 \times 4 \times 256 \times 256$, stride $= 2$ |
| ReLu |
| Conv $4 \times 4 \times 256 \times 256$, stride $= 2$ |
| ReLu |
| FC $4096 \times 256$ |
| ReLu |
| FC $256 \times 256$ |
| ReLu |
| FC $256 \times 2mD$ |

**Evaluation metrics**. For evaluating the compositional generalization, we follow Xu et al. (2022) to use the implementations[4] in scikit-learn (version 0.22) for the linear models of compositional generalization metrics. Specifically, we use the function `linear_model.LogisticRegressionCV` with default settings for classification, and use function `linear_model.RidgeCV` with alphas=$\{0, 0.01, 0.1, 1.0, 10\}$ for regression. There are $N = 500$ samples for training models of the metrics. We use the same metrics configurations as in (Locatello et al., 2019) to evaluate the disentanglement performance. Since the representation used is vector-valued and can not be evaluated directly, we thus follow (Yang et al., 2022; Du et al., 2021) to perform PCA as post-processing on the representation and evaluate the performance with these metrics.

---

[1] https://github.com/xrenaa/DisCo
[2] https://github.com/ThomasMrY/VCT
[3] code is in https://openreview.net/forum?id=ue4CArRAsct
[4] https://github.com/wildphoton/Compositional-Generalization

Table 4: Decoder $\mathcal{D}$ architecture used in vec-VAE-based methods.

| |
|---|
| FC $mD \times 256$ |
| ReLu |
| FC $256 \times 256$ |
| ReLu |
| FC $256 \times 4096$ |
| ConvTranspose $4 \times 4 \times 256 \times 256$, stride $= 2$ |
| ReLu |
| ConvTranspose $4 \times 4 \times 256 \times 256$, stride $= 2$ |
| ReLu |
| ConvTranspose $4 \times 4 \times 256 \times 128$, stride $= 2$ |
| ReLu |
| ConvTranspose $4 \times 4 \times 128 \times 64$, stride $= 2$ |
| ReLu |
| ConvTranspose $7 \times 7 \times 64 \times 3$, stride $= 2$ |

Table 5: MLP in vector-valued models. (a) We use $m$ MLPs in vec-VCT. (b) We change the output of SAE's MLP from $m$ to $mD$. (c) We change the input of SAE's Str-Tfm layer from $1$ to $D$.

| Encoder MLP | Decoder MLP |
|---|---|
| FC $256 \times 128$ | FC $D \times 128$ |
| ReLu | ReLu |
| FC $128 \times D$ | FC $128 \times 256$ |

(a) MLP in vec-VCT

| |
|---|
| FC $256 \times 256$ |
| ReLu |
| FC $256 \times 256$ |
| ReLu |
| FC $256 \times mD$ |

(b) MLP in SAE encoder

| |
|---|
| FC $D \times 64$ |
| ReLu |
| FC $64 \times 128$ |
| ReLu |
| FC $128 \times (64 \times 2)$ |

(c) MLP in Str-Tfm

**Computational cost**. No matter vector-valued or scalar-valued methods, we use an Nvidia V100 16G as the compute resource for conducting the experiments.

**Reproducibility**. Our code will be released upon acceptance.

## E   MORE RESULTS ON METRICS

The main paper only presents the important experiment results to support the key findings. We provide more results with disentanglement metrics $\beta$-VAE score and MIG. In order to analyze the influence of the training-testing splitting ratio, we provide more results with different ratios.

**Results of more metrics**. Table 6 presents the results of additional disentanglement metrics in this section: $\beta$-VAE score, MIG. Consistently to our results in the main paper, we observe that the vec-models have comparable or even better performance on some metrics. The results further support that the vec-model can simultaneously process the compositional generalization and disentanglement.

**Results of different training-testing splitting ratio** In the main paper, we follow (Xu et al., 2022) to conduct the experiments mainly with a training-testing splitting ratio (split ratio) of $1 : 9$. In this section, we also provide the results of the experiments with three different ratios: $3 : 7$, $1 : 9$, and $5 : 95$. As shown in Figure 6, the split ratio has limited influence on the compositional metrics and disentanglement metrics, which is also observed in (Xu et al., 2022). However, their experiments are conducted on scalar-valued methods.

## F   MORE RESULTS ON REGULARIZATION STRENGTH

In this section, we provide results that regularization strength is less than five as the supplement results. We also provide the results on the MPI3D dataset to present more support for our findings.

Table 6: Comparisons of more disentanglement metrics between the scalar-valued and vector-valued methods (mean ± std, higher is better). For vec-VCT*, $D = 256$ is the same as Yang et al. (2022).

| Method | Shapes3D | | MPI3D | |
|---|---|---|---|---|
| | $\beta$-VAE | MIG | $\beta$-VAE | MIG |
| *Scalar-valued:* | | | | |
| FactorVAE | $0.89 \pm 0.05$ | $0.21 \pm 0.12$ | $0.51 \pm 0.05$ | $0.11 \pm 0.05$ |
| $\beta$-TCVAE | $0.88 \pm 0.07$ | $0.37 \pm 0.19$ | $0.46 \pm 0.03$ | $0.11 \pm 0.05$ |
| SAE | $0.99 \pm 0.01$ | $0.32 \pm 0.10$ | $0.79 \pm 0.03$ | $0.18 \pm 0.06$ |
| VCT | $0.97 \pm 0.05$ | $0.40 \pm 0.11$ | $0.78 \pm 0.05$ | $0.26 \pm 0.05$ |
| *Vector-valued:* | | | | |
| vec-FactorVAE | $0.98 \pm 0.02$ | $0.25 \pm 0.07$ | $0.46 \pm 0.07$ | $0.10 \pm 0.05$ |
| vec-$\beta$-TCVAE | $0.94 \pm 0.04$ | $0.12 \pm 0.07$ | $0.49 \pm 0.04$ | $0.04 \pm 0.01$ |
| vec-SAE | $0.96 \pm 0.05$ | $0.22 \pm 0.09$ | $0.70 \pm 0.06$ | $0.15 \pm 0.09$ |
| vec-VCT | $0.99 \pm 0.02$ | $0.42 \pm 0.10$ | $0.77 \pm 0.05$ | $0.35 \pm 0.07$ |
| vec-VCT* | $1.00 \pm 0.00$ | $0.44 \pm 0.08$ | $0.74 \pm 0.05$ | $0.33 \pm 0.06$ |

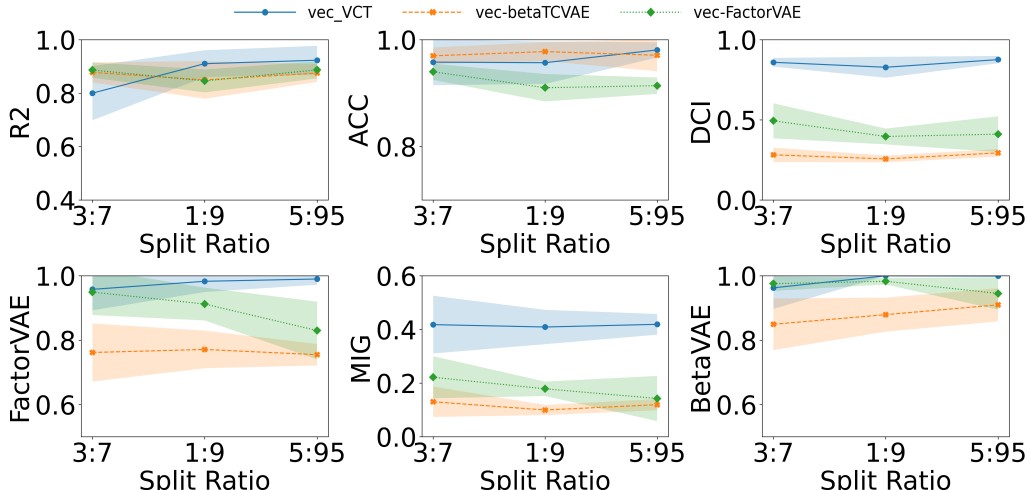

Figure 6: Generalization and disentanglement performance vs training-testing splitting ratio on Shapes3D. Three vector-valued methods (with vector size $D = 64$, regualarization strength $\gamma = 10.0$) are evaluated. vec-VCT, vec-betaTCVAE, and vec-FactorVAE use varying training-testing splitting ratios $\{3 : 7, 1 : 9, 5 : 95\}$.

**FactorVAE when $\gamma \leq 5$.** Figure 8 shows the results of vec-FactorVAE and vec-$\beta$-TCVAE when the regularization strength $\gamma \in \{0.5, 1, 2, 4\}$. We can observe that the results support the explanation in the main paper: if the regularization strength is small for vec-FactorVAE, there also is a significant improvement in the compositional generalization and disentanglement performance as the increase of $\gamma$. We also observe that the phenomenon is consistent between vec-FactorVAE and vec-$\beta$-TCVAE.

**Results on MPI3D.** To further support our conclusion on the regularization strength $\gamma$ of vector-valued methods, we conduct the same experiment on MPI3D. As shown in Figure 11, the results are consistent with our results in the main paper. We observe that as the regularization strength increase, both the compositional generalization and the disentanglement drops for vec-FactorVAE but improves for vec-$\beta$-TCVAE. The additional metrics (MIG and $\beta$-VAE score) are consistent with other metrics.

**Pearson correlation coefficient on MPI3D.** In the main paper, the experiments reveal the relation between compositional generalization and disentanglement based on the results of Shapes3D. We also provide such evidence on MPI3D here. From the results in Figure 9, we observe consistent patterns: the classification performance is positively correlated to the disentanglement, but there is no significant correlation between the regression and disentanglement performance.

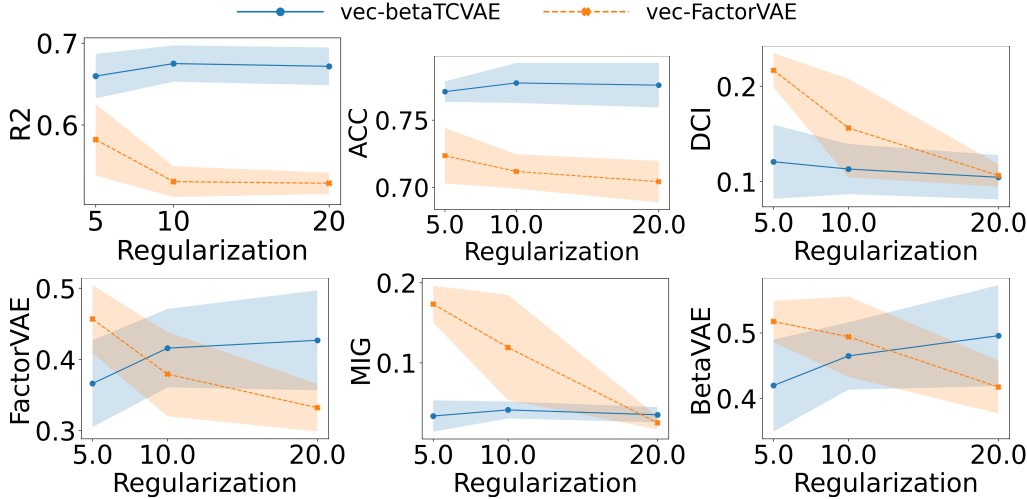

Figure 7: Generalization and disentanglement performance vs regularization strength on MPI3D. Two vector-valued methods (with vector size $D = 64$) are evaluated. vec-betaTCVAE and vec-FactorVAE use varying regularization strength $\gamma \in \{5, 10, 20\}$.

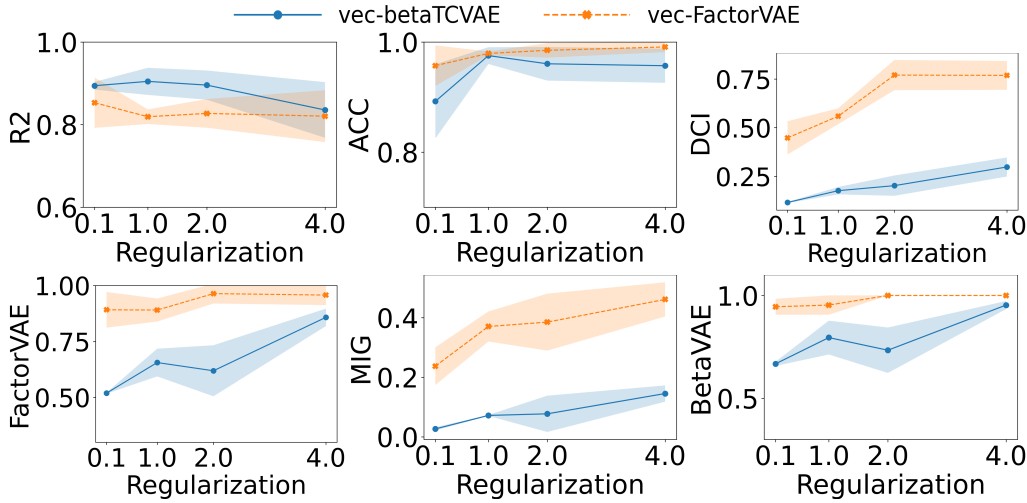

Figure 8: Generalization and disentanglement performance vs regularization strength on Shapes3D. Two vector-valued methods (with vector size $D = 64$) are evaluated. vec-betaTCVAE and vec-FactorVAE use varying regularization strength $\gamma \in \{0.1, 1.0, 2.0, 4.0\}$.

## G   MORE RESULTS ON VECTOR SIZE

In this section, we provide results that vector size between 2 and 32 as the supplement results for the main paper. We also provide the results on the MPI3D dataset.

**More results on Shapes3D**. In the main text, we only considered some specific and important vector dimension values, which are $\{1, 2, 32, 64\}$. However, due to the large span between 2 and 32, one may be interested in other values within the interval. Therefore, we also experimented with the values within the interval. Figure 10 shows the results when vector size is $\{4, 8, 16, 24\}$. We also observe that as the increase vector dimension (size) increased, both the compositional and the disentanglement consistently improved, which provides further evidence for the conclusion in Section 6.2.

**Results on MPI3D**. In order to further support our conclusion on vector size of vector-valued methods, we conduct the same experiment on MPI3D. As shown in Figure 11, we see that as the vector dimension (size) increases, the compositional generalization performance of the models improves

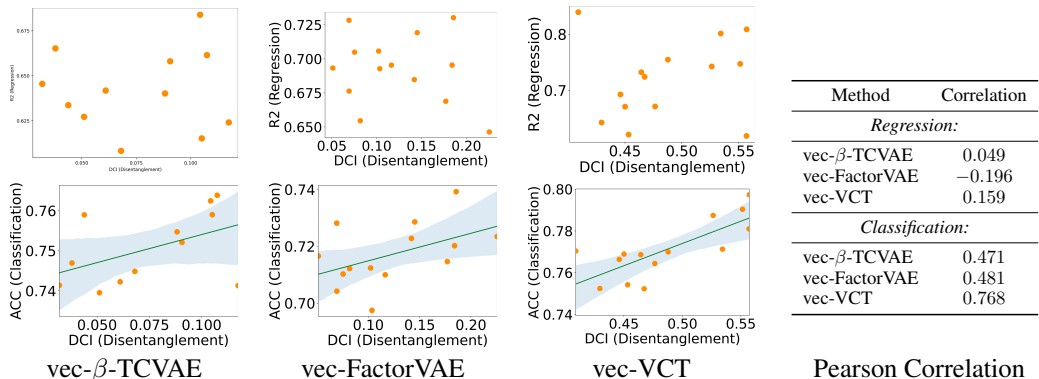

Figure 9: Generalization performance vs disentanglement performance on the MPI3D with $D = 64$ and $\gamma = 10$. The Pearson correlation coefficient is calculated in the table. Orange data points represent instances of the same vector-valued method trained using different random seeds.

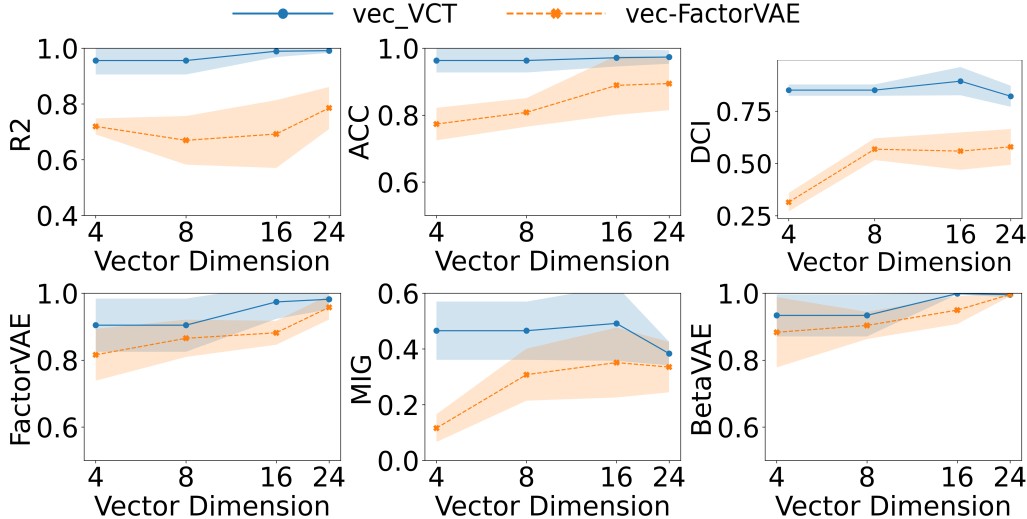

Figure 10: Generalization and disentanglement performance vs regularization strength on Shapes3D. Two vector-valued methods (with regularization strength $\gamma = 10$) are evaluated. vec-betaTCVAE and vec-FactorVAE use varying vector size $D \in \{4, 8, 16, 24\}$.

consistently. The disentanglement performance of certain models (e.g., VCT) also shows improvement. However, some models do not significantly exhibit such improvements in some of the metrics. The reason may be that these methods have poor performance on these metrics on MPI3D. Consequently, changing the vector dimension does not result in substantial performance changes for these models.

## H   LIMITATION OF OUR STUDY

Our study builds upon prior work Xu et al. (2022), in which experiments were conducted on the two proposed metrics. To the best of our knowledge, these metrics are the only ones directly evaluating compositional generalization with random train-test splitting. Although our experiments were carried out on synthetic (Shapes3D) and realistic (MPI3D) datasets, the factors within these datasets are relatively simple. Furthermore, the number of factors is known and relatively small. While we have demonstrated a set of models with strong disentanglement and generalization performance, the potential applications of these models still need to be explored.

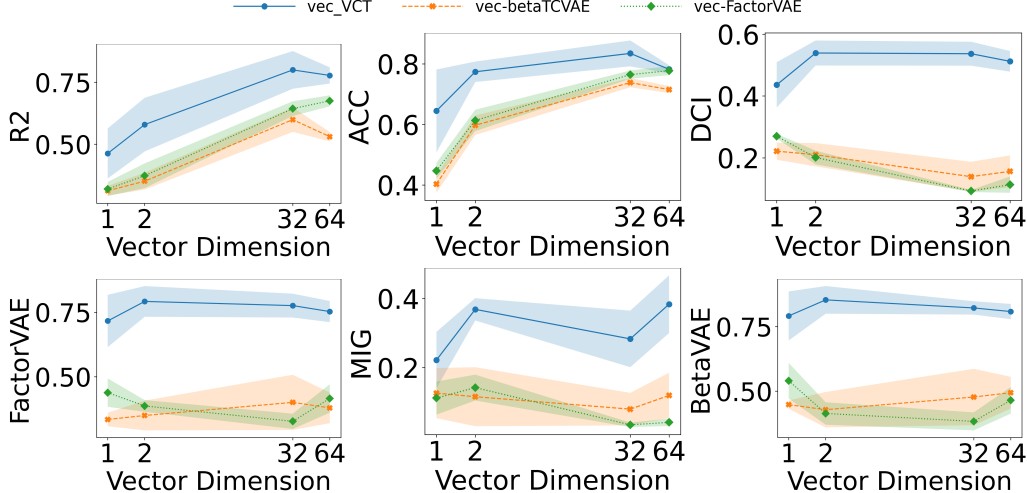

Figure 11: Generalization and disentanglement performance vs regularization strength on MPI3D. Two vector-valued methods (with regularization strength $\gamma = 10$) are evaluated. vec-betaTCVAE and vec-FactorVAE use varying vector size $D \in \{1, 2, 32, 64\}$.

Table 7: Directly increase the representation evaluation experiment. We evaluate the following ideal and learned representation in terms of compositional generalization and disentanglement.

| Method | R2 | ACC | DCI |
|---|---|---|---|
| *scalar-valued:* | | | |
| Ideal | 1.00 | 0.99 | 0.99 |
| Shifted | 0.46 | 0.75 | 0.99 |
| Matrix | 0.99 | 1.00 | 0.18 |
| Matrix + shifted | 0.00 | 0.45 | 0.20 |
| *vector-valued:* | | | |
| Ideal | 1.00 | 0.99 | 0.99 |
| Shifted | 0.38 | 0.50 | 0.99 |
| Matrix | 0.99 | 1.00 | 0.17 |
| Matrix + shifted | 0.00 | 0.45 | 0.17 |

(a) Ideal representation

| Method | $R^2$ | ACC | DCI |
|---|---|---|---|
| *scalar-valued:* | | | |
| $\beta$-TCVAE | 0.30 | 0.56 | 0.61 |
| FactorVAE | 0.49 | 0.48 | 0.50 |
| *vector-valued:* | | | |
| $\beta$-TCVAE embed | 0.30 | 0.56 | 0.61 |
| $\beta$-TCVAE repeat | 0.30 | 0.56 | 0.61 |
| FactorVAE embed | 0.49 | 0.48 | 0.50 |
| FactorVAE repeat | 0.49 | 0.48 | 0.50 |

(b) Learned representation

# I   MORE SANITY CHECKS

We first assess the representation derived from ground truth values, which is ideally a perfect representation. This representation is referred to as the ideal representation.

We seek to determine if scalar-valued representations can achieve high performance on both types of metrics. Given that ground truth values are scalar, the ideal representation can be constructed as follows: $(i)$ constructing the ideal scalar-valued representation by directly normalizing the ground truth values of corresponding factors, and $(ii)$ creating the ideal vector-valued representation by normalizing the ground truth values of factors and multiplying the normalized values with a random embedding, which increases the vector size of the corresponding factor. This embedding is shared across the training and testing set. As shown in Tab. 7 (a), both vector-valued and scalar-valued representations perform optimally on compositional generalization and disentanglement.

We employ two methods to corrupt the disentanglement and generalization abilities of ideal representations: $(i)$ applying different linear operations on the training and testing set ($y = \alpha x + \beta$, where $\alpha, \beta$ are scalars) to the ideal representation to corrupt generalization ability, and $(ii)$ multiplying the ideal representation with a random invertible matrix to entangle the representation of different factors, thus corrupting disentanglement ability. In Tab. 7 (a), no significant performance improvement is observed for vector-valued representation.

Although the aforementioned corruption provides insights, there remains a gap between learned representations and artificially corrupted representations. To examine this in a non-ideal setting, we

propose two ways to directly map scalar-valued representations to vector-valued ones: $(i)$ multiplying the learned scalar-valued representation of VAE with a random embedding of the corresponding factor, and $(ii)$ directly repeating the scalar of the learned representation of VAE. Tab. 7 (b) uses $\beta$-TCVAE and FactorVAE as examples, showing no significant improvement in vector-valued representation metrics compared to the original scalar-valued one.

**Implications** The performance gain is not solely due to directly increasing the representation dimension. We also verified that perfect generalization and disentanglement performance can be obtained with an ideal scalar representation.

