# OpenReview forum: "Vector-valued Representation is the Key: A Study on Disentanglement and Compositional Generalization"
_ICLR.cc/2024/Conference — ICLR 2024 Conference Withdrawn Submission_

### Official Review · Reviewer_9DDo · 2023-10-26

**Soundness:** 1 poor
**Presentation:** 3 good
**Contribution:** 2 fair
**Rating:** 5
**Confidence:** 2

**Summary:**

This paper investigates the hypothesis that using vector (rather than scalar) representations for disentangled latent variables improves both disentanglement and compositional generalization. It introduces vectorized variants of three existing methods for disentangled representation learning (beta-TCVAE, FactorVAE, and SAE) and also a de-vectorized version of VCT. The results show improved accuracy across the board for vectorized versions of the methods.

Note: I previously reviewed this an earlier version manuscript for another conference.

**Strengths:**

The paper investigates an interesting hypothesis and proposes a number of new methods as part of its investigation. It conducts experiments on 2 different datasets which allows a minimal assessment of generalizability, and it also evaluates more than one metric for disentanglement as well as compositional generalization. The results are striking.

**Weaknesses:**

The method of vectorizing an existing method seems to be based on ad-hoc heuristics and not particularly generalizable. In the case of beta-TCVAE, vectorization amounts having components of a vector being constrained to the same variance, and for the total correlation to be evaluated within each dimension. These modifications do not seem very principled and are more like heuristics. In the case of SAE, it seems like one simply increases dimensionality by a factor, although I could be wrong since I'm not familiar with the SAE method.

The paper includes experiments claiming to demonstrate the difference between vectorization and increased dimensionality. These show that for an "ideal" representation, both vectorized and scalar methods perform well. But this observation fails to demonstrate the difference between vectorized methods and increased-dimensionality scalar methods in non-ideal settings. A more straightforward comparison between vectorized methods and matched-dimension scalar methods would be more conclusive.

**Questions:**

* How does "vec-SAE" differ from SAE with more dimensions?
* Can you give a more systematic set of principles for vectorizing a VAE-type method, that would help the reader to vectorize a method not covered by your paper?

---

> ### Author Response · Authors · 2023-11-20
> **Rebuttal by Authors-Part1**
>
> Thank you for your valuable comments and suggestions. We appreciate your feedback and have made several changes to address your concerns. Please find below our responses to each point you raised.
> - The estimation of vector-based total correlation is a very interesting point to research on, although the method is heuristic one, but the method is indeed a good choice for vec-betaTCVAE. Moreover, it's important to note that the total correlation estimation of vec-betaTCVAE method itself is **not** the **primary contribution** of our work. Inspired by your comment, we have attempted the following three different methods, and have added these parts into (Section 6.4 and Appendix C highlighted in blue). Note that estimating the total correlation (TC) in vector-valued representations is inherently a difficult problem. We propose three additional ways to approximate TC. These methods are described below, and the experimental results demonstrate that our method performs better in practice than these alternatives.
>
>   1. The total correlation directly derived from multidimensional Gaussian distribution. The specific derivation and calculation methods can be found in the Appendix C of the paper, referred to as "vec-betaTCVAE Sum".
>   2. For each training step, we randomly select a variable for each representation vector to calculate TC, referred to as "vec-betaTCVAE Rand".
>   3. We calculate TC as the average of multiple runs of the method 2, referred to as "vec-betaTCVAE Rand Sum"
> (See Appendix C for details)
>
>   The performance of these methods is shown in the table below. Based on our results, our current vectorization method still performs the best, despite the issue raised by the reviewer. We hypothesize that the potential reason for this might be some latent inductive bias in the network, which makes our method even more effective.
>
> | Method                     | Factor           | DCI             | R^2            | ACC            |
> |----------------------------|------------------|-----------------|----------------|----------------|
> | vec-β-TCVAE Sum            | 0.58 ± 0.14      | 0.22 ± 0.09     | 0.79 ± 0.09    | 0.73 ± 0.21    |
> | vec-β-TCVAE Rand           | 0.79 ± 0.08      | 0.31 ± 0.07     | 0.88 ± 0.05    | 0.96 ± 0.02    |
> | vec-β-TCVAE Rand Sum       | 0.52 ± 0.09      | 0.32 ± 0.10     | 0.70 ± 0.07    | 0.51 ± 0.04    |
> | vec-β-TCVAE (Ours)         | 0.82 ± 0.08      | 0.31 ± 0.08     | 0.87 ± 0.05    | 0.98 ± 0.01    |
>
>
> For SAE, please refers to Q1.

---

> > ### Comment · Reviewer_9DDo · 2023-11-21
> > **Overall response**
> >
> > Thanks for the additional explanations and experiments.  I highly encourage the inclusion of these additions to the main text of the paper, since they will greatly improve the applicability of the paper to those who want to vectorize new methods beyond those of the paper.  While I still have questions, for now I am tentatively raising my score to 5.

---

> ### Author Response · Authors · 2023-11-21
> **Rebuttal by Authors-Part2**
>
> - We are regret that our writting make you misunderstanding that our ideal experiments are to show such difference. We have modicate this part in Section 6.5 (highlight in blue). We would like to highlight: we conduct the ideal experiments to show that in the most optimal situations (i.e., our designed ideal cases), scalar methods can also achieve excellent disentanglement and combinatorial generalization capabilities. Furthermore, we wanted to investigate whether the metrics used in our study prefer vector representations. Our experiments showed that directly mapping scalar-value representations to vector-value based ones did not lead to an increase in CG metrics.
>
>    On the other hand, your suggestion to compare vectorized methods and purely increased dimensionality from a more direct perspective is well received. In response to this, we have conducted additional experiments where the total dimensions were consistent across scalar-based and vector-based methods. The results are as follows:
>
> | Method           | Shapes3D - FactorVAE | Shapes3D - DCI | Shapes3D - R2 | Shapes3D - ACC | MPI3D - FactorVAE | MPI3D - DCI | MPI3D - R2 | MPI3D - ACC |
> |------------------|----------------------|----------------|---------------|----------------|-------------------|-------------|------------|-------------|
> | FactorVAE        | 0.83 ± 0.06          | 0.44 ± 0.12    | 0.46 ± 0.18   | 0.39 ± 0.10     | 0.31 ± 0.04        | 0.21 ± 0.01 | 0.30 ± 0.02 | 0.39 ± 0.02 |
> | β-TCVAE         | 0.83 ± 0.10          | 0.65 ± 0.16    | 0.45 ± 0.15   | 0.47 ± 0.18     | 0.44 ± 0.05        | 0.27 ± 0.01 | 0.32 ± 0.03 | 0.45 ± 0.03 |
> | SAE              | 0.98 ± 0.04          | 0.87 ± 0.12    | 0.72 ± 0.05   | 0.90 ± 0.17     | 0.71 ± 0.04        | 0.47 ± 0.05 | 0.55 ± 0.07 | 0.77 ± 0.02 |
> | VCT              | 0.95 ± 0.05          | 0.86 ± 0.02    | 0.56 ± 0.24   | 0.58 ± 0.15     | 0.72 ± 0.04        | 0.47 ± 0.03 | 0.39 ± 0.13 | 0.69 ± 0.09 |
> | FactorVAE (640)  | 0.77 ± 0.05          | 0.56 ± 0.12    | 0.77 ± 0.10   | 0.65 ± 0.10     | 0.46 ± 0.03        | 0.43 ± 0.01 | 0.37 ± 0.03 | 0.49 ± 0.02 |
> | β-TCVAE (640)   | 0.81 ± 0.11          | 0.74 ± 0.10    | 0.59 ± 0.15   | 0.76 ± 0.18     | 0.43 ± 0.02        | 0.39 ± 0.01 | 0.35 ± 0.02 | 0.44 ± 0.02 |
> | vec-FactorVAE    | 0.93 ± 0.06          | 0.55 ± 0.11    | 0.88 ± 0.05   | 0.96 ± 0.02     | 0.38 ± 0.06        | 0.16 ± 0.05 | 0.53 ± 0.02 | 0.71 ± 0.01 |
> | vec-β-TCVAE  | 0.82 ± 0.08          | 0.31 ± 0.08    | 0.87 ± 0.05   | 0.98 ± 0.01    | 0.42 ± 0.06        | 0.11 ± 0.03 | 0.67 ± 0.02 | 0.78 ± 0.01 |
> | vec-SAE       | 0.89 ± 0.08          | 0.63 ± 0.06    | 0.95 ± 0.01   | 0.98 ± 0.01    | 0.62 ± 0.08        | 0.33 ± 0.09 | 0.87 ± 0.03 | 0.88 ± 0.01 |
> | vec-VCT       | 0.98 ± 0.04          | 0.85 ± 0.06    | 0.91 ± 0.10   | 0.80 ± 0.09    | 0.70 ± 0.06        | 0.48 ± 0.04 | 0.70 ± 0.07 | 0.77 ± 0.02 |
> | vec-VCT*      | 0.97 ± 0.04          | 0.89 ± 0.02    | 0.99 ± 0.02   | 0.90 ± 0.03    | 0.66 ± 0.03        | 0.45 ± 0.06 | 0.85 ± 0.07 | 0.78 ± 0.02 |
>
> Please note that FactorVAE (640) has the same total dimension with vec-FactorVAE and β-TCVAE (640) has the same total dimension with vec-β-TCVAE. Our results indicate that under these conditions of consistent latent dimensionality, our compositional generalization still outperforms the new baseline. This might be due to the fact that the disentanglement learning of VAEs is ill-posed when the dimensionality is too large, resulting in poor performance on both disentanglement and compositional generalization.

---

> > ### Comment · Reviewer_9DDo · 2023-11-21
> > **Thanks for these experiments**
> >
> > Thank you for conducting these experiments.  Is it surprising that vec-beta-TCVAE has similar FactorVAE score to beta-TCVAE in both Shapes3d and MPI3d?  Or that vec-SAE has lower FactorVAE than SAE in MPI3d?

---

> ### Author Response · Authors · 2023-11-21
> **Rebuttal by Authors-Part3**
>
> Q1: In our paper, the definition of scalar-value methods is that a factor is encoded by a single dimension, while for vector-value methods, a factor is encoded by several dimensions. SAE utilizes a decoder structure where each layer encodes a unique factor. Consequently, for scalar-value methods, each layer of the decoder is modulated by a single dimension. In contrast, in vector-value method, each layer is modulated by several dimensions.
>
> Therefore, to increase the dimensions in scalar-value methods, we would need to increase the number of layers in the decoder. For each added dimension, an additional layer is required because, by definition, each layer of a scalar-based SAE encodes only one factor. If we were to increase the number of dimensions per layer, the method would transition into a vector-valued method.
>
> Q2: Thank you for your valuable suggestion. To vectorize a method not covered in our paper, we suggest the following systematic set of principles:
>   1. Identify the basic structure of the model that learns a single factor. Then, extend this basic structure from one dimension to multiple dimensions.
>   2. Confirm that the inductive bias for disentanglement remains valid after the extension.
>   3. If the inductive bias still holds after the extension, then we have successfully vectorized the method. If it does not, we need to explore how to modify the inductive bias so that it remains valid for vector representations.

---

> > ### Comment · Reviewer_9DDo · 2023-11-21
> > **More questions**
> >
> > Thanks for addressing these questions.
> >
> > > Identify the basic structure of the model that learns a single factor. Then, extend this basic structure from one dimension to multiple dimensions.
> >
> > Could you clarify what you mean by "basic structure"?

---

> ### Author Response · Authors · 2023-11-21
> **Response to Reviewer 9DDo**
>
> Dear Reviewer 9DDo,
>
> We would like to thank you for follow-up and increasing the score. We appreciate your valuable feedback and will continue to enhance our work. We fully agree with you that these additions greatly improve the applicability and also sync them into the main text.
> - by "basic structure", we mean the structure or loss function (or regularization term) that plays the role of inductive bias.
> For example, "basic structure" learns a single factor, including: 1. the input of AdaIN-like structure in SAE. 2. each output token in VCT. 3. $q_\phi(z_i)$ in total correlations $KL(q_\phi(z)|\prod_i^m q_\phi(z_i))$.
> We would like to change "basic structure" into "loss function/architecture inductive bias for disentanglement" for clarity instead. Also, we will include these examples.
> - If we misunderstand, please tell us, we are more than happy to answer any remaining questions.
>   1. Is it the numbers correct? We checked the code for collecting results and the printed numbers of our experiment and found that the numbers in the Table are correct.
>   2. Is it in line with the expectation that vec-beta-TCVAE and vec-SAE do not outperform their scalar-based version on FactorVAE score? Yes, the total correlation used in vec-beta-TCVAE is a heuristic approximation, so the disentanglement performance is expected to be worse than the scalar-valued one with a good estimation. SAE does not leverage any regularization but only the structure itself. There is no explicit constraint on mutual information between the layers. Therefore, the disentanglement performance of vec-SAE is expected to be worse than the scalar-valued SAE with a smaller information bottleneck (limited by scalar).

---

> ### Author Response · Authors · 2023-11-23
>
> Thank you for all of your constructive comments and suggestions. We have added the principles to vectorize new methods except those of the paper in conclusion. Please let us know if you have any further questions or concerns, since the discussion stage is coming to an end in thirty minutes.

---

### Official Review · Reviewer_czbp · 2023-10-31

**Soundness:** 3 good
**Presentation:** 2 fair
**Contribution:** 2 fair
**Rating:** 6
**Confidence:** 5

**Summary:**

The paper explores the relation between disentanglement and compositional generalisation when each factor value is encoded using vector-valued representations as opposed to scalar-values ones (as is common in previous work). To this end the paper expands the definition of standard disentanglement penalties (used for training) and metrics (for evaluation) to the vector-value case. They then use apply them to a handful of models in order to probe their disentanglement and generalisation properties. The authors find that when using said vector-valued there is a positive correlation between disentanglement (as measured using several metrics) and the generalisation capabilities across all models. This highlighting the importance of distributed representations on the generalisation ability of Deep Learning models.

**Strengths:**

The idea of expanding the study of disentanglement to the vector based cased is a natural next step from previous work that uses scalar valued representations. The main hypothesis is thus clear. The description of how the different penalties and metrics are extended is also easy to understand. Finally, the results support the conclusion that the authors reach, i.e. that there is a positive correlation between disentanglement and generalisation when using vector-values representations.

**Weaknesses:**

In spite of the above there are some issues that need addressing before I can recommend publication. In no particular order:

1. There are some important references that the authors have missed: Schott et al. (2021), and Montero et al., (2022) for example goes beyond the VAE-based disentanglement studies of Montero 2021, though they are still restricted to the scalar-values case. The authors also cite Singh et al., (2022) but don't mention that this reference talks about compositional generalisation and vector valued representations. This is the most relevant reference for this study, yet there is no discussion about the relation between the two.
2. Section 2.2 is a bit confusing and needs a rewrite. As currently written, it is stated that disentanglement has not been considered when studying generative models, but the authors themselves stated before that both Montero et al. and Xu et al. study both in the VAE-based setting, a generative model. So which one is it? Also, Xu is not the only one to use random splits. Schott et al., also dies this.
3. The structure based disentanglement needs much more details to be understandable. The short explanation and the figure are not nearly enough to understand why this is an interesting model to test.
4. The authors claim to use the beta-VAE and MIG metrics but they never appear in the table.

**Questions:**

1. What are metrics like accuracy, r2 applied to? The full vector-values representation? What are the targets?
2. The implications paragraph in section 6.3 is completely unclear to me.
3. Relatedly, I personally I believe that DCI is the most accurate disentanglement measure, so it is striking that vector-values representation are not providing any improvement and this severely undermines the conclusions of the paper, especially since the authors don't say how the are applying the other metrics.

---

> ### Author Response · Authors · 2023-11-20
> **Rebuttal by Authors-part1**
>
> Thank you for your valuable comments and suggestions. We appreciate your feedback and have made several changes to address your concerns. Please find below our responses to each point you raised.
> - Thank you for bringing to our attention the omission of the references in our previous work description. We appreciate your insight and have made the necessary changes as per your suggestion (highlight in blue in Section 2). The revised descriptions for the three papers are as follows:
>    1. Schott et al. (2021); Montero et al. (2021) go beyond the VAE-based disentanglement studies, though they are still restricted to the scalar-valued representation.
>     2. Singh et al. also emphasize the importance of vector-valued representation to compositional generalization.
> - Thank you for your detailed feedback and for pointing out the areas of confusion in Section 2.2 of our paper. We are regret  that our writing has led to your misunderstanding. We have revised the writtings in Section 2.2, which is highligted in blue. When we mentioned that disentanglement has not been considered in the study of generative models, we were specifically referring to the work by Zhao et al. (2018). We regret that if our wording implied that Xu was the only one who used this approach. Our intention was to emphasize his work along with Montero's but not mentioed that they are the only one. We value your input and will make these modifications in our revised version to avoid these missunderstading.
> - We understand your concern about the level of detail provided and we apologize if it was insufficient for a thorough understanding. We appreciate your suggestion to elaborate on why these methods are interesting for testing. We have revised this section in the paper (highlighted in blue in Section 3.2 & 3.3) to provide a clearer and more detailed explanation. We believe that this additional context will help readers better understand our choice of the method for testing.
> - We would like to clarify that the results obtained from the use of Beta-VAE and MIG metrics are indeed included in our appendix E.  We apologize for any confusion this may have caused, we have highlighted in the main text that the rest of the results are in appendix E.

---

> ### Author Response · Authors · 2023-11-20
> **Rebuttal by Authors-part2**
>
> Q1: The metrics such as accuracy and R2 are indeed applied to the full vector-valued representation in our study. The primary aim of these metrics is to verify the model's ability to generalize to new combinations. In other words, it is meant to assess how well the model can identify the values of these factors under new data combinations.
>
> Q2: We are regret that our writing caused your confusion in Section 6.3. We have revisited and revised the section for clarity. The main content of this section is: In the context of vector-valued representation, we observe a strong positive correlation between the metric of classification and disentanglement, while the metric of regression does not show such a strong correlation. We speculate that the underlying reason might be that regression primarily occurs in factors with multiple values, such as azimuth with 15 values on Shapes3D (For classification, such as scale with 3 values). This leads that the representation of this factor will encode more information. The size of the vector dimensions significantly impacts R2 in such cases.
>
> Q3: We appreciate your insights regarding the strengths and other potential weaknesses of our work. We  acknowledge that  DCI indeed is also a significant metric. However, we wish to clarify that we do not appreciate to base our conclusions solely on a single metric. In our paper, we have employed multiple measures to evaluate the performance, in line with the widely accepted standards prevalent in the scholarly community. The approach to refer to multiple standards to assess such capabilities has indeed become a consensus within the academic world [1,2,3,4]. On the other hand, our primary focus in the paper is to explore the relationship between disentanglement and generalization. We propose that there exist vector-valued methods that can possess excellent capabilities in both disentanglement and generalization. By vectorizing the methods, we observe an improvement in the combinatorial generalization ability. However, we did not claim that vectorization is a universal technique for improving disentanglement capabilities.
>
> [1] Learning disentangled representation by exploiting pretrained generative models: A contrastive learning view. In International Conference on Learning Representations, 2021.
>
> [2] Structure by architecture: Structured representations without regularization. In The Eleventh International Conference on Learning Representations, 2022.
>
> [3] Infodiffusion: Representation learning using information maximizing diffusion models. international conference on machine learning, 2023.
>
> [4] Visual concepts tokenization. Advances in Neural Information Processing Systems, 2022.

---

> > ### Comment · Reviewer_czbp · 2023-11-23
> >
> > I thank the authors for their extensive rebuttal. Most of my concerns have been addressed and I am improving my score.

---

> > > ### Author Response · Authors · 2023-11-23
> > >
> > > Dear Reviewer czbp,
> > >
> > > Thank you for your positive feedback and for raising the score. We truly appreciate the time and effort you have dedicated to reviewing our work.

---

### Official Review · Reviewer_2XW7 · 2023-11-01

**Soundness:** 3 good
**Presentation:** 3 good
**Contribution:** 3 good
**Rating:** 6
**Confidence:** 4

**Summary:**

The authors study the challenge of compositional generalization and its relationship to disentanglement. Specifically, they propose that leveraging ideas of vector-valued disentanglement has the potential to improve the performance of disentangled representation learning methods on compositional generalization tasks. They introduce multiple methods in order to extend existing scalar-valued models of disentanglement to their vector-valued setting, and demonstrate that indeed these models appear to yield greater accuracy on a test of compositional generalization. They further study the correlation between disentanglement and compositional generalization, showing that there appears to be a relationship between the two tasks for classification metrics, but not for regression metrics.

----

### Post Rebuttal Edit
We thank the authors for their extensive response to our review and for addressing our major concerns. We greatly appreciate their inclusion of dimensionality matching models, and believe this has significantly improved the quality of the paper. For this reason we have raised our score. We would ultimately like to see all models with matching dimensionality, regardless of their number of parameters since it is likely that the number of parameters has a significantly lesser influence on disentanglement than latent dimensionality. Equivalently the authors may increase the number of parameters in their vector valued models in order to equate the two on other terms. However, at this stage we still think the comparison for FactorVAEs and Beta TCVAEs is valuable on its own and therefore can be satisfied to weakly recommend acceptance of the work. We furthermore appreciate the authors increasing the clarity of their notation and for updating their related work.

**Strengths:**

- The paper is relatively well written and easy to read.
- The topic of vector-valued disentanglement is very interesting and under-explored. Specifically, the relationship to compositional generalization is also interesting, and if their findings can be shown to hold, this would have a significant impact on the field.
- Performance on compositional generalization is very high and there is a clear significant difference between vector valued and scalar valued models on the two datasets examined (although the methodological procedure raises some doubts, see weaknesses).

**Weaknesses:**

- It appears that one of the main experimental findings of the paper (the fact that vector valued representations are correlated with improved compositional generalization) may be conflated with the simple increase in dimensionality resulting from increasing the dimension of the latent space. Specifically, in Table 1, it appears the authors did not maintain a comparable latent dimensionality between their scalar-valued and vector-valued models. (For example, it currently appears all vector valued models have a latent dimensionality which is D-times bigger than the scalar valued baselines). As shown by the authors in Figure 2, this simple increase in dimensionality yields dramatic improvements to compositionality metrics even for the vanilla baseline Vec-AE. The authors attempt to address this concern in Section 6.4, Table 3, however they only compare 2 models on a single dataset, and furthermore do not include standard deviations (!). Unfortunately, given this is one of the main results of the paper, I believe this is insufficient. All scalar-valued baselines for Table 1 should have consistent total dimensionality with the vector valued counterparts, especially considering this is shown to have a significant impact on performance. I would ask the authors to include these results in a rebuttal in order to have faith in the main claims of the paper.

- The extension of scalar valued models to vector valued versions is a bit confusing and ill-defined, as outlined below:
	- Equation (5) is confusing due to potential overload of indexing notation. It would be helpful if the authors put bounds on their sums in this case so that it is clear exactly which vector is being referred to by z_j. Is this the j’th vector (of dimension D)? Or a new vector composed of the j’th dimensions of all other vectors (E.g. something like [z_{0,j}, z_{1,j}, z_{2,j}, …, z_{m,j}])?
	- If I understand equation (5) correctly, the new Total Correlation loss is really only seeking to minimise the element-wise total correlation of the individual vectors, meaning for example that the second dimension of the first vector (z_{1,2}) and the first dimension of the second vector (z_{2,1}) have no penalty on their correlation. Similarly, for the dimensions within a vector, there is no constraint on correlation. Is this true? If so, this seems like a quite significant departure from the original TCVAE idea. It would be appreciated if the authors could provide greater motivation for why this is a good disentanglement loss in their model.
	- The derivation of Equation (5) in the appendix is not given. The authors derive why the original approximation is no longer fit, but then simply propose the new total correlation loss without much discussion. Further justification of this loss should be included. If the authors could comment on that here, that would be greatly appreciated.
	- A more detailed description of all vector valued methods should be provided (for example the Vec-FactorVAE should be described entirely in the appendix).

- Some references to related work are missing. Notably, there is a line of work related to the discovery of vector-valued directions in latent space which correspond to disentangled transformations [1,2,3]. It would be helpful if the authors could address this line of work as well, as I believe it is at least related conceptually.

- The authors provide little to no intuition for why a vector valued representation may achieve compositional generalization better than a scalar value counterpart.

- In Table 1, the authors bold their values (for FactorVAE metric and DCI) which are not significantly above the baseline scalar-valued models. This is potentially misleading, and the authors should either bold all values that are not statistically significantly different, or they should remove the bolding.

[1] Christos Tzelepis, Georgios Tzimiropoulos, and Ioannis Patras. WarpedGANSpace: Finding non-linear rbf paths in GAN latent space. In ICCV, 2021

[2] Yue Song, Andy Keller, Nicu Sebe, and Max Welling. Flow Factorized Representation Learning. In NeurIPS. 2023.

[3] Yuxiang Wei, Yupeng Shi, Xiao Liu, Zhilong Ji, Yuan Gao, Zhongqin Wu, and Wangmeng Zuo. Orthogonal jacobian regularization for unsupervised disentanglement in image generation. In ICCV, 2021

**Questions:**

- What is the marginal q(z_j) in equation (5) referring to?
- Can the authors provide a compelling reason for why they did not match the dimensionality of the baseline scalar-valued models and vector-valued models in Table 1?

---

> ### Author Response · Authors · 2023-11-20
> **Rebuttal by Authors-part1**
>
> Thank you for your valuable comments and suggestions. We appreciate your feedback and have made several changes to address your concerns. Please find below our responses to each point you raised.
>
> - We appreciate the reviewer for raising this point, and we fully understand his concern. This indeed might be a question that most readers would have. Therefore, we have added the following experiments, which include results of the two types of VAEs on two datasets regarding disentanglement and compositional generalization across four metrics. Our results indicate that under these conditions of consistent latent dimensionality, our compositional generalization still outperforms the new baseline. This might be due to the fact that the disentanglement learning of VAEs is ill-posed when the dimensionality is too large, resulting in poor performance on compositional generalization. Please note that we have not conducted this experiment for methods like SAE and VCT for the following reasons:
>   1. For structural-based methods like VCT and SAE, the number of representation units is related to the number of network layers (SAE) and the size of features (VCT). More representation units lead to a significant increase in the number of parameters, which would result in unfair comparisons.
>   2. If we use too many representation units, the computational cost of the model would be quite high. We do not have sufficient computational resources to conduct these set of experiments.
> | Method           | Shapes3D - FactorVAE | Shapes3D - DCI | Shapes3D - R2 | Shapes3D - ACC | MPI3D - FactorVAE | MPI3D - DCI | MPI3D - R2 | MPI3D - ACC |
> |------------------|----------------------|----------------|---------------|----------------|-------------------|-------------|------------|-------------|
> | FactorVAE        | 0.83 ± 0.06          | 0.44 ± 0.12    | 0.46 ± 0.18   | 0.39 ± 0.10     | 0.31 ± 0.04        | 0.21 ± 0.01 | 0.30 ± 0.02 | 0.39 ± 0.02 |
> | β-TCVAE         | 0.83 ± 0.10          | 0.65 ± 0.16    | 0.45 ± 0.15   | 0.47 ± 0.18     | 0.44 ± 0.05        | 0.27 ± 0.01 | 0.32 ± 0.03 | 0.45 ± 0.03 |
> | SAE              | 0.98 ± 0.04          | 0.87 ± 0.12    | 0.72 ± 0.05   | 0.90 ± 0.17     | 0.71 ± 0.04        | 0.47 ± 0.05 | 0.55 ± 0.07 | 0.77 ± 0.02 |
> | VCT              | 0.95 ± 0.05          | 0.86 ± 0.02    | 0.56 ± 0.24   | 0.58 ± 0.15     | 0.72 ± 0.04        | 0.47 ± 0.03 | 0.39 ± 0.13 | 0.69 ± 0.09 |
> | FactorVAE (640)  | 0.77 ± 0.05          | 0.56 ± 0.12    | 0.77 ± 0.10   | 0.65 ± 0.10     | 0.46 ± 0.03        | 0.43 ± 0.01 | 0.37 ± 0.03 | 0.49 ± 0.02 |
> | β-TCVAE (640)   | 0.81 ± 0.11          | 0.74 ± 0.10    | 0.59 ± 0.15   | 0.76 ± 0.18     | 0.43 ± 0.02        | 0.39 ± 0.01 | 0.35 ± 0.02 | 0.44 ± 0.02 |
> | vec-FactorVAE    | 0.93 ± 0.06          | 0.55 ± 0.11    | 0.88 ± 0.05   | 0.96 ± 0.02     | 0.38 ± 0.06        | 0.16 ± 0.05 | 0.53 ± 0.02 | 0.71 ± 0.01 |
> | vec-β-TCVAE  | 0.82 ± 0.08          | 0.31 ± 0.08    | 0.87 ± 0.05   | 0.98 ± 0.01    | 0.42 ± 0.06        | 0.11 ± 0.03 | 0.67 ± 0.02 | 0.78 ± 0.01 |
> | vec-SAE       | 0.89 ± 0.08          | 0.63 ± 0.06    | 0.95 ± 0.01   | 0.98 ± 0.01    | 0.62 ± 0.08        | 0.33 ± 0.09 | 0.87 ± 0.03 | 0.88 ± 0.01 |
> | vec-VCT       | 0.98 ± 0.04          | 0.85 ± 0.06    | 0.91 ± 0.10   | 0.80 ± 0.09    | 0.70 ± 0.06        | 0.48 ± 0.04 | 0.70 ± 0.07 | 0.77 ± 0.02 |
> | vec-VCT*      | 0.97 ± 0.04          | 0.89 ± 0.02    | 0.99 ± 0.02   | 0.90 ± 0.03    | 0.66 ± 0.03        | 0.45 ± 0.06 | 0.85 ± 0.07 | 0.78 ± 0.02 |
>
> 2.1 We regret that the notation here is not clear enough, we have modified the text in the paper (highlighted in blue). z_j represents the j’th dimensions of all vectors.

---

> ### Author Response · Authors · 2023-11-20
> **Rebuttal by Authors-part2**
>
> - 2.2 & 2.3 You are correct in your understanding that our new Total Correlation loss does not impose a penalty on the correlation between different dimensions of different vectors, for instance, between $z_{1,2}$ and $z_{2,1}$. We understand that this might seem counterintuitive, however, our following empirical results speak to the effectiveness of this approach, which suggests that our model still works well even without explicit constraints on correlations of these terms. As for Equation (5),  it is a heuristic method. Although we also have actually obtained another three vectorized versions, but found that it did not outperform the orginal version of vec-betaTCVAE in practice. The reasons for this are not entirely clear and require further investigation.
>
>     We think this is a very interesting point and we appreciate the reviewer's suggestion. Inspired by your comment, we have attempted the following three different methods, and have added these parts into our paper (Section 6.4 and Appendix C highlighted in blue). Note that estimating the total correlation (TC) in vector-valued representations is inherently a difficult problem. We propose three additional ways to approximate TC. These methods are described below, and the experimental results demonstrate that our method performs better in practice than these alternatives.
>   1. The total correlation directly derived from multidimensional Gaussian distribution. The specific derivation and calculation methods can be found in the Appendix C of the paper, referred to as "vec-betaTCVAE Sum".
>   2. For each training step, we randomly select a variable for each representation vector to calculate TC, referred to as "vec-betaTCVAE Rand".
>   3. We calculate TC as the average of multiple runs of the method 2, referred to as "vec-betaTCVAE Rand Sum"
> (See Appendix C for details)
>
>   The performance of these methods is shown in the table below. Based on our results, our current vectorization method still performs the best, despite the issue raised by the reviewer. We hypothesize that the potential reason for this might be some latent inductive bias in the network, which makes our method even more effective.
>
> | Method                     | Factor           | DCI             | R2            | ACC            |
> |----------------------------|------------------|-----------------|----------------|----------------|
> | vec-β-TCVAE Sum            | 0.58 ± 0.14      | 0.22 ± 0.09     | 0.79 ± 0.09    | 0.73 ± 0.21    |
> | vec-β-TCVAE Rand           | 0.79 ± 0.08      | 0.31 ± 0.07     | 0.88 ± 0.05    | 0.96 ± 0.02    |
> | vec-β-TCVAE Rand Sum       | 0.52 ± 0.09      | 0.32 ± 0.10     | 0.70 ± 0.07    | 0.51 ± 0.04    |
> | vec-β-TCVAE (Ours)         | 0.82 ± 0.08      | 0.31 ± 0.08     | 0.87 ± 0.05    | 0.98 ± 0.01    |
>
> -  2.4.Thank you for your suggestion, we have add the details of vec-FactorVAE model in the appendix B.
>
> - 3.We appreciate the reviewer bringing these works to our attention. We have now addressed these studies in our paper (highlight in blue) as follows: there is a considerable body of work [1,2,3] exploring how to identify the semantic directions in their latent space, and these directions are precisely represented by vectors. These representations are vector-valued one for semantics.
>
> - 4.The underlying intuition is that essential differences arise between a single-dimensional space and a multi-dimensional one. For instance, the skew lines in three-dimensional space does not exist in a two-dimensional plane. The orthogonality of random vectors in higher-dimensional spaces does not exist in low dimensional space . Therefore, by expanding the single-dimensional representation of a factor in our model to a multi-dimensional one, we hypothesize that the model may be able to learn some fundamental differences. This multi-dimensionality could potentially contribute to achieving better compositional generalization than a scalar one.
>
> - 5.Thank you for your valuable feedback. We have opted to remove the bolding from the values in Table 1 and  it was not our intention to mislead our readers.

---

> ### Author Response · Authors · 2023-11-20
> **Rebuttal by Authors-part3**
>
> Q1: Thank you for your inquiry regarding equation (5). As depicted in our new Figure in Appendix C, the term marginal $q(z_j)$ refers to the joint probability distribution of the $j$-th dimension of all the vector representations. We have clarified this point in our paper.
>
> Q2: Thank you for your question regarding the dimensionality matching of our baseline scalar-valued models and vector-valued models in Table 1. We considered two main factors in making this decision. However, in light of your suggestion, we have now included cases with equal total dimensions in Table 1 to provide readers with a more comprehensive understanding.
> From the perspective of disentangled representation learning, each latent unit is intended to learn a single factor. If the total dimensions were equal, scalar models would have more encoding units in the latent space, leading to an unfair comparison.
> We took into account models such as SAE and VCT, where the number of model parameters is directly proportional to the number of latent units. More latent units would result in a larger number of parameters for these models, again leading to an unfair comparison.

---

### Official Review · Reviewer_RASM · 2023-11-05

**Soundness:** 3 good
**Presentation:** 3 good
**Contribution:** 3 good
**Rating:** 6
**Confidence:** 5

**Summary:**

The paper investigates the relationship between disentanglement and compositional generalization when varying the dimension of factors of variation (i.e., scalar and vector valued factors).

**Strengths:**

- The paper is very well written and easy to understand.
- The paper performs thorough experiments to understand how employing scalar/vector valued representations effect disentanglement and compositional generalization capabilities.
- The experiments showing the role of bottleneck size (in the case of vector valued representations) and how it effects disentanglement and generalization capabilities are very helpful.

**Weaknesses:**

None as such.

**Questions:**

- Ways to improve approximation of total correlation for vector valued representations could be further helpful to improve the results of the paper.
- it will be helpful to see if the learned vector valued representations can further boosts downstream results in case of visual reasoning problems or RL problems.
- The key message behind the paper (vector valued representations are important for compositional generalization) had already been argued in previous work (like RIMs [1] or its followers, or Discrete Key-Value Bottleneck [2]).

[1] RIMs, https://arxiv.org/abs/1909.10893

[2] Discrete Key Value Bottleneck, https://arxiv.org/abs/2207.11240

---

> ### Author Response · Authors · 2023-11-20
> **Rebuttal by Authors**
>
> Thank you for your valuable comments and suggestions and the encouraging words. We appreciate the time and effort you took to review our paper. Please find below our responses to your concerns and the changes we have made to address them.
>
> - The way of estiamting the total correlation is a very interesting point and we appreciate the reviewer's suggestion. Inspired by your comment, we have attempted the following three different methods, and have added these parts into our paper (Section 6.4 and Appendix C highlighted in blue). Note that estimating the total correlation (TC) in vector-valued representations is inherently a difficult problem. We propose three additional ways to approximate TC. These methods are described below, and the experimental results demonstrate that our method performs better in practice than these alternatives.
>
>   1. The total correlation directly derived from multidimensional Gaussian distribution. The specific derivation and calculation methods can be found in the Appendix C of the paper, referred to as "vec-betaTCVAE Sum".
>   2. For each training step, we randomly select a variable for each representation vector to calculate TC, referred to as "vec-betaTCVAE Rand".
>   3. We calculate TC as the average of multiple runs of the method 2, referred to as "vec-betaTCVAE Rand Sum"
> (See Appendix C for details)
>
>
>
>
> | Method                     | Factor           | DCI             | R^2            | ACC            |
> |----------------------------|------------------|-----------------|----------------|----------------|
> | vec-β-TCVAE Sum            | 0.58 ± 0.14      | 0.22 ± 0.09     | 0.79 ± 0.09    | 0.73 ± 0.21    |
> | vec-β-TCVAE Rand           | 0.79 ± 0.08      | 0.31 ± 0.07     | 0.88 ± 0.05    | 0.96 ± 0.02    |
> | vec-β-TCVAE Rand Sum       | 0.52 ± 0.09      | 0.32 ± 0.10     | 0.70 ± 0.07    | 0.51 ± 0.04    |
> | vec-β-TCVAE (Ours)         | 0.82 ± 0.08      | 0.31 ± 0.08     | 0.87 ± 0.05    | 0.98 ± 0.01    |
>
> The performance of these methods is shown in the table below. Based on our results, our current vectorization method still performs the best. We hypothesize that the potential reason for this might be some latent inductive bias in the network, which makes our method even more effective.
>
> - Thank you for your insightful comment on investigating the applicability of our learned vector-valued representations to enhance downstream tasks.  We conduct experiments  in visual reasoning and unfairness. We believe that such an experiment could indeed provide a valuable perspective to our work. We will included these additional experiments in the appendix of our paper and make references to them in the main text. We utilized the settings from disentanglement_lib to evaluate our representations. The experiment results will be updated here.
>
> - Thank you for pointing out the two papers in the field of generalization (Recurrent Independent Mechanisms (RIMs) [1] and Discrete Key-Value Bottleneck [2]). We have discuss them in the updated version.  While both these papers emphasize the importance of vector-valued representations for generalization, our work differs significantly in its primary focus. Our research is specifically centered on the aspect of compositional generalization, which is a distinct and critical point of study within the broader context of generalization. Moreover, our key claim is that vector-valued representation is the key simultaneous presence of both disentanglement and compositional generalization abilities, a key feature that we believe sets our work apart from the cited papers. While the cited works have made important strides in understanding the role of vectorization in generalization, our research delves deeper into the relationship between vectorization and compositional generalization, which is also an important perspective.

---

### Comment · Area_Chair_Tbk5 · 2023-11-20
**Discussion between authors and reviewers**

Dear Reviewers,

Thanks for the reviews. The authors have uploaded their responses to your comments, please check if the rebuttal address your concerns and if you have further questions/comments to discuss with the authors. If the authors have addressed your concerns, please adjust your rating accordingly or vice versa.

AC

---

### Author Response · Authors · 2023-11-21

We thank all the reviewers for the insightful feedback and constructive comments, as well as their encouraging words: the paper is very (relatively) **well written** (Reviewer RASM, 2XW7) and **easy to understand** (Reviewer RASM, 2XW7, czbp),  performs **thorough** experiments, The experiments are **very helpful**. Weaknesses: **None as such** (Reviewer RASM). vector-valued disentanglement is **very interesting** and **under-explored** (Reviewer 2XW7, 9DDo). The relationship to compositional generalization is also **interesting**. (Reviewer 2XW7) and if their findings hold, this would have a **significant impact** on the field. (Reviewer 2XW7); Performance on compositional generalization is **very high**, the results are **striking** (Reviewer 2XW7, 9DDo), The idea is a **natural next step** from previous work (Reviewer czbp). The main hypothesis is clear (Reviewer czbp). The results support the conclusion (Reviewer czbp). This **highlighting** the importance of distributed representations on the generalization ability of Deep Learning models. (Reviewer czbp). Proposes a number of new methods (Reviewer 9DDo).

The improvements in the main draft can be listed as (highlighted in blue):
1. As suggested by Reviewers RASM, 2XW7 and czbp, we revised the related work section and add some interesting related works.
2. As suggested by Reviewers 9DDo, czbp, 2XW7, we add more description on SAE and VCT in Section 3.2 and 3.3, espetially on why we choose these two models.
3. As suggested by Reviewers 2XW7, we improve the notations in Eq. 5 to avoid misunderstanding.
4. As suggested by Reviewers 2XW7 and 9DDo, we conduct new experiments with consistent total dimensionality and add the results in Table 1.
5. As suggested by Reviewers RASM, 2XW7, 9DDo, we explored different methds for estimation of TC for vec-$\beta$-TCVAE and add the results with Section 6.4 and add descriptions of the new estimation methods in Appendix C.
6. As suggested by Reviewer RASM, we add the details of the vec-FactorVAE in Appendix B.
7. As suggested by Reviewer 9DDo，we add the principles to vectorize new methods except those of the paper in conclusion.

---

### Author Response · Authors · 2023-11-23

Thank you for all of your constructive comments and suggestions. Please let us know as soon as possible if you have any further questions or concerns, since the discussion stage is coming to an end soon.